

# TPVTrack v1.0: A watershed segmentation and overlap correspondence method for tracking tropopause polar vortices

Nicholas Szapiro[1] and Steven Cavallo[1]

[1]School of Meteorology, University of Oklahoma, Norman, OK, USA

**Correspondence:** Nicholas Szapiro (nick.szapiro@ou.edu)

**Abstract.** A new modular, free software package is described for tracking tropopause polar vortices (TPVs) natively on structured or unstructured grids. Motivated by limitations in spatial characterization and time tracking within existing approaches, TPVTrack leverages the dynamics of TPVs to represent their (1) spatial structure, with variable shapes and intensities and (2) time evolution, with mergers and splits. TPVs are segmented from the gridded flow field into spatial objects as restricted regional watershed basins on the tropopause, described by geometric metrics, associated over time by overlap similarity into major and minor correspondences, and tracked along major correspondences. Simplified segmentation and correspondence test cases illustrate some of the appeal, sensitivities, and limitations of TPVTrack, including effective representation of spatial shape and reduced false positive associations in time. Tracked TPVs in more realistic historical conditions are consistent in bulk with expectations of life cycle and mean structure. Individual tracks are less reliable when discriminating between multiple overlaps. Modifications to track other physical features are possible, with each application requiring evaluation.

## 1 Introduction

Among the disturbances on the extratropical tropopause, tropopause polar vortices (TPVs) are common, coherent upper-level potential vorticity anomalies with typical radii of 100 to 1000 km and lifetimes of days to months (Hakim, 2000; Hakim and Canavan, 2005). TPV-like features play documented roles in so-called potential vorticity thinking (e.g., Hoskins et al. 1985; Holton and Hakim 2013), surface cyclogenesis and high-impact weather (e.g., Davis and Emanuel 1991; Morgan and Nielsen-Gammon 1998; Grams et al. 2011; Simmonds and Rudeva 2012), geographic climate (e.g., Shapiro et al. 1987; Nieto et al. 2008; Kew et al. 2010), stratosphere-troposphere exchange (e.g., Sprenger et al. 2007), potential predictability (e.g., Provenzale 1999), and evaluation of model skill (e.g., Béguin et al. 2012). Knowledge of the structure and history of TPVs may aid further systematic and reproducible approaches in these and more areas. For example, TPV tracks can aid a mechanistic understanding of how distant radiosonde observations can impact the development of a surface cyclone days later (Yamazaki et al., 2015). Accurate, automated tracking of TPVs in gridded data improves our knowledge of TPV structure and history. Within a framework of Ertel's potential vorticity (Rossby, 1939; Hoskins et al., 1985; Pedlosky, 2013) in a composite sense (Cavallo and Hakim, 2010), dynamics of TPVs are dominated by quasi-horizontal advection subject to generally smaller diabatic and frictional forcings. Diagnostic trajectories and prognosed scalar transport further support the advection-dominated dynamics for individual cases (not shown).



Tracking approaches for various features follow an approach of spatial identification and time correspondence (Hodges, 1999; Limbach et al., 2012; Neu et al., 2013; Ullrich and Zarzycki, 2017). However, choices and details within and between approaches can lead to systematic differences and uncertainties in results, both from algorithm sensitivities and the dynamics of the feature. For tropical cyclones, sensitivities to thresholds for size, wind speeds, warm cores, genesis, and duration impact

track statistics (Walsh et al., 2007; Horn et al., 2014). For extratropical surface cyclones, deeper and stronger features are more consistent and robust across methods (Neu et al., 2013; Tilinina et al., 2013). Differences in tracks across algorithms, especially for features without independent or consensus definitions like TPVs, complicate use and interpretation as each approach is perfect within its own circular definition (Neu et al., 2013). Because our primary interest is in the physical TPVs rather than the tracked objects, clarity in the characteristics of each algorithm may lend confidence and rationale for informing whether

results derived from automated tracks are artifacts or physical, especially if an ensemble with complementary approaches can be constructed by opportunity (e.g., Tebaldi and Knutti (2007)) or design.

Towards such a usable description of TPVTrack, Sect. 2 frames TPV tracking approaches and describes the current algorithm design and implementation. Idealized segmentation and correspondence test cases with reference truths quantify some of the appeal, sensitivities, and limitations of components of TPVTrack, including relative to several existing approaches (Sect. 3.1).

Individual and aggregated historical cases relate the tracked objects to physical expectations of TPVs (Sect. 3.2). Section 4 concludes with a summary and possible extensions to other features.

## 2  TPV Tracking

Synoptically, TPVs are coherent anomalies on the dynamic tropopause associated with the larger polar vortex that have a regional minimum in potential temperature and cyclonic circulation or regional maximum in potential temperature and anticy-

clonic circulation that last over time. Defining "coherent," "regional," "associated," and "last" are fundamental to an automated scheme. Note that the existence of a local extremum in a continuous surface implies the existence of a closed contour about that extremum, thus consistent with a notion of vortices (rather than waves) with trapped fluid for frictionless and adiabatic flow (Hakim and Canavan, 2005).

### 2.1  Existing algorithms

We summarize the automated approaches we know have been used to track and characterize TPVs or similar features in terms of multiple steps: choice and pre-processing of gridded data, segmentation in space, characterization of features, and tracking in time. Several limitations of existing approaches motivated the design and implementation of TPVTrack. With further details in the references, here we focus on comparing the interdependent choices that define "regional," "coherent," and "lasts." Defining "associated" with the larger polar vortex is typically a post-processing step and applicable to any set of tracks, e.g., Hakim and

Canavan (2005) subset an "Arctic" category of tracks that last at least 2 days with over 60% of their lifetimes north of 65° N.

Each method bases objects about extrema. Since a surface can have many local extrema, to target salient features, filtering criteria are imposed so not all local extrema are cores of objects; an extremum must be "regional." For Hakim and Canavan

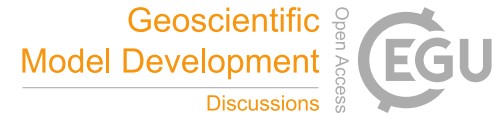



(2005), using 2.5° data, any local extremum must also be an extremum within a 650 km radius. For Kew et al. (2010), background potential vorticity is calculated by averaging over a large ($10^7$ km$^2$) area. Positive anomalies are candidate extrema, and a flow-dependent contour of smaller prescribed area ($1.7 \times 10^5$ km$^2$) is computed about each candidate. Lesser maxima within the bounding contour of a stronger maximum are disregarded. Simmonds and Rudeva (2014) require a local extremum in the surface Laplacian and minimum concavity for saliency through a threshold (unspecified) on an area-averaged Laplacian. In effect, choices for saliency and neighborhoods are imposed filters, integral to each approach and with a number of trade-offs as is typical of filter design.

What area is associated or "coherent" with a regional extremum? Spatially, where is the edge of the feature? For Hakim and Canavan (2005), eight spokes are extended from the vortex core until the radial gradient in potential temperature changes sign for each. The minimum of these potential temperature values across all eight radials defines the bounding isentrope. For Kew et al. (2010), the flow-dependent contour of prescribed area also defines the feature in space. If overlap occurs between two features, the weaker anomaly shrinks to a smaller contour yielding no overlap. For Simmonds and Rudeva (2014), radius is the weighted mean distance from the core to where the Laplacian vanishes. Consider how these definitions would represent two TPVs, one physically larger than the other. For Kew et al. (2010), the area is prescribed and the same for both TPVs. For the local gradient-based methods, sensitivity to smaller-scale noise in the surface can trigger stopping criteria with resulting areas somewhat arbitrary. Note that for some approaches (e.g., point methods) spatial shape is simply an additional diagnostic, whereas other approaches may leverage the additional information (e.g., for overlap tracking).

When is a TPV the same feature as at the previous time? How long does an individual TPV last? Hakim and Canavan (2005) use a simple proximity algorithm where a track is extended if a vortex core is within 600 km of the previous location (advected previous location for later versions). Corresponding to a maximum vortex speed of 28 m s$^{-1}$, the threshold distance subjectively works for the intended polar region and also is easily reproduced. Kew et al. (2010) develop an approach akin to contour surgery, where cells are advected forward in time by analyzed winds, and the resulting shape is from a convex hull of the destinations after exclusion of outlier locations. Significant (10%) spatial overlap extends a track. Note that both of these extension strategies duplicate the entire history of a TPV if there are multiple matches. Simmonds and Rudeva (2014) estimate a new position of each cyclone by weighting the track's past and climatological velocities, calculate probability of associations based on distance and central pressure differences, and keep the combination of associations within candidate groups (constructed by linking all positions within a cutoff distance) that maximize the product of pairwise probabilities as tracks. As reflected in the summary above, associations in time are decisions for whether the expected evolution sufficiently matches the actual states. Defining an expected evolution can involve a range of assumptions and complexity impacting fidelity, but approaches may be related notionally. For example, using a prescribed search radius between point locations is equivalent to calculating horizontal overlap for features with prescribed shapes of disks and no transport. Comparisons are illustrated more concretely in Sect. 3.1.



## 2.2 Description of TPVTrack

An implementation in Python 2.7 of this algorithm supporting and tested with output from the National Center for Environmental Prediction's Global Forecast System (GFS; e.g., Saha et al. (2010)), European Center for Medium Range Weather Forecasting interim reanalysis (ERA-Interim; Dee et al. (2011)), Weather Research and Forecasting Model (WRF; Skamarock

et al. (2005)), and atmospheric component of the Model for Prediction across Scales (MPAS-A; Skamarock et al. (2012)) through a modular, object-oriented approach is publicly available (Sect. 4). A user's guide is included with the package. The following outlines the core modules in TPVTrack. Each module outputs one NetCDF (Rew and Davis, 1990; Whitaker, 2015) file.

### 2.2.1 Input data

The meteorological inputs to the algorithm are gridded zonal wind, meridional wind, relative vertical vorticity, and potential temperature on the extratropical tropopause (Gettelman et al., 2011; Ivanova, 2013) over time. With no diabatic or frictional forcings, an air parcel would have fixed potential vorticity and potential temperature. With small forcings for the associated parcels, TPVs can be tracked as material eddies on the dynamic tropopause (with preferred diagnosis described in Sect. 3.2.3).

A subset of the original domain ($[-\frac{\pi}{2}, \frac{\pi}{2}] \times [0, 2\pi)$ in radians latitude/longitude) may be specified as the domain of interest.

Within the domain, values are interpreted in a finite volume sense as areal means located at cell centers. In deriving the tropopause surface, some data sources have missing values (e.g., GFS uses a missing value if the entire column is above the potential vorticity threshold). If no additional information is available, we fill missing values in pre-processing by an iterative extrapolation from valid nearest neighbors. We do not track globally since a potential vorticity isosurface does not correspond to the tropopause near the tropics (e.g., Highwood and Hoskins (1998)). The default is to track polewards of a user-specified

latitude, set to $\pm 30°$ for the northern and southern hemispheres, respectively, as the average position of the subtropical jet stream to be inclusive of candidate TPVs. Especially since we are interested in polar regions, we treat the longitudinally duplicated north and south poles in latitude/longitude grids specially to be consistent with the finite volume interpretation throughout the rest of the grid. Each physical pole is treated as a single polar cap with the pole neighboring number of longitude points .

To handle various data sources modularly, each is supported natively by Mesh and Cell classes defining the geometry and topology of the domain. The Cell class provides methods for obtaining the location, area, nearest neighboring cells, and cells within a specified radius of a cell in the mesh. For example, the left neighbor of a cell in a latitude/longitude mesh with index (iLat, iLon) is (iLat, (iLon-1) mod nLon). The Mesh class provides methods for the latitudes, longitudes, and areas of all cells and finding the closest cell to a point. This allows for optimizations within a unified code implementation. For

example, consider finding the cell center closest to a point in a latitude/longitude grid versus an MPAS Voronoi unstructured mesh. For a latitude/longitude grid, we can leverage the underlying structure to simply find the closest latitude and longitude independently. For an unstructured mesh, the provided connectivity of cell-to-cell neighbors can be used. Starting from a guess cell, the closest cell to the point can be found by iteratively walking towards a closer neighbor until no closer neighbor is



found. Both approaches involve fewer operations than a brute force global minimum of all cells' distances to the point. Various meshes could be unified through a common background data structure (e.g., spatial quadtree), but this is not required.

### 2.2.2 Spatial segmentation

Segmentation maps the mesh cells to disjoint spatial objects at each time (Fig. 1). Model tropopause data contain signals on
a spectrum of wavelengths, from planetary to grid scales. These scales come from both the originating model and the post-processing that interpolates to the tropopause. As illustrated in Sect. 3.1.1, robustly segmenting TPV-scale objects requires some filtering of smaller scales. Two key questions are: what counts as an extremum? What area is associated to an extremum?

A value is a local minimum if none of the values of its nearest neighbors are smaller. Local maxima can be identified by searching for local minima on the negative of the surface. There are currently two options to define a regional minimum within a
prescribed region (e.g., 300 km radius disk): the smallest value or the smallest local minimum. While qualifying as the smallest value requires a comparison to all other values in the region, qualifying as the smallest local minimum is a comparison only to other local minima. A minimum across the region is a stronger filter that will result in fewer objects, while only comparing to other minima permits distinct objects as long as the minima are sufficiently separate. This may be beneficial for a tropopause with undulations within a significant larger-scale gradient. The distinction vanishes for a surface with numerous local minima.
The prescribed area is a fundamental user-defined filter for a scale of saliency. Alternately, basins with less than a minimum amplitude could be filtered by merging into neighbors. This option has not been explored, since it precludes identifying weak amplitude TPVs, and a reference potential temperature is needed.

For the area coherent with an extremum, local growing methods suffer from a noisy surface triggering stopping criteria (e.g., Sect. 3.1.1), so a broader perspective is necessary to be more robust. Considering the tropopause as a surface with local
undulations, watershed basins naturally represent the spatial objects. A watershed basin is defined by a minimum and a basin that drains by steepest descent to the minimum. Particularly given the grid scale undulations, a direct watershed transform of the surface results in a classical over-segmentation (Serra, 1983) as each local minimum forms a separate object. To resolve the over-segmentation produced by small scale noise, we implemented a variation of a marker-controlled watershed transform with the regional minima as markers.
The regional watershed basins are formed in two steps. First, each cell is associated to a local minimum by following a path of local steepest descent through nearest neighbors. In case of a plateau, the cell with the smaller global index is chosen for consensus. While the occurrence of bit-equal values is rare in general, lossy data compression, idealized conditions, and other systematic factors can increase the likelihood of plateaus. Second, to form basins only associated with regional minima, each local minimum that is not a regional minimum is redirected to drain into the most intense minimum in the neighborhood,
iterating until each local minimum drains into a regional minimum.

In order to segment a surface into cyclonic and anticyclonic TPVs, we first calculate watershed basins from the potential temperature surface (lows) and the negative of the potential temperature surface (highs), so each cell is initially mapped to two basins. Then, all regional extrema are mapped to themselves, and each non-extremum cell is mapped to its corresponding either high or low basin by the sign of its local relative vorticity (Fig. 1.b). Thus, an anticyclonic non-extremum cell is mapped

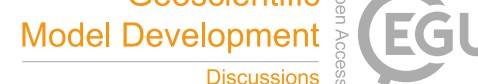



to its high regional extremum. Note that – since highs and lows are assigned based on vorticity – objects are disjoint, but each individual basin is not necessarily connected. While connectivity can be imposed by, say, flood-filling from the core of each vortex, doing so can remove filamentary sub-structures possibly worth preserving.

Since the resulting segmentation is driven by steepest descent of the input surface, the segmentation is sensitive to undulations in the foothills away from a vortex core. To reduce this sensitivity and better isolate the anomalies, optionally, a bounding contour can further restrict each basin (Fig. 1.c). The cells outside the bound are mapped to a background. We identify boundary cells within each basin as cells with a neighbor not in the basin. A user-defined percentile (e.g., $10^{th}$) of boundary cell potential temperature amplitudes with respect to the core then sets the contour independently for each basin. A last closed contour approach ($0^{th}$ percentile) is noticeably less robust to outliers causing severe over-restriction, especially as the originating mesh spacing decreases (Sect. 3.2.1). Restricting the basins tends to produce smaller, more consistent objects over time. This consistency is appealing both physically and algorithmically to improve subsequent correspondence and tracking.

### 2.2.3 Shape metrics

The flow field associated with a TPV is expected to be related to the TPV's shape (e.g., Thorpe 1986; Masarik and Schubert 2013). Principally, a given idealized potential vorticity anomaly is partitioned between static stability and vorticity; a broad and shallow feature has more stability, while one that is narrow and deep has more vorticity. Moreover, the spatially larger an anomaly, the stronger the induced circulation (Hoskins et al., 1985). Towards building physical understanding and connections, various properties of a basin's geometry may be quantified by using the segmentation map of cells to objects as masks.

We organize these metrics by dimensionality: point, boundary, and area metrics. Point metrics include the latitude, longitude, and potential temperature of the core and maximum amplitude (maximum minus minimum potential temperature in the basin). Boundary cells are identified as cells with a nearest neighbor not in the same basin. Metrics include circularity (length of the perimeter with respect to the length of the perimeter of a circle with equivalent area), eccentricity (ratio of fit major and minor axes), boundary potential temperature (median of the potential temperatures of boundary cells) and boundary amplitude (difference of the boundary potential temperature from the the potential temperature of the core). Area metrics are area-weighted integrals over the basin, including area, equivalent radius (of a planar disk with equivalent area), mean vorticity, and circulation. Incorporating a vertical dimension to a TPV's shape is not straightforward. A TPV's vertical extent does not extend throughout the entire column. Extending a mask on the tropopause to nearby vertical levels would need to account for the possibility of vertically tilted flow.

### 2.2.4 Time correspondence

Correspondences associate spatial objects between consecutive times. Motivated by the advection-dominated dynamics, the first step is to identify possible correspondences using similarity measured by "horizontal" plus "vertical" overlap. As depicted in Fig. 2, consider candidate basins A at time $t_0$ and B at time $t_0 + \Delta t$.

Horizontal overlap is calculated by advecting A forward $\Delta t/2$ and B backward $-\Delta t/2$ in time on a sphere using the local winds at $t_0$ and $t_0 + \Delta t$, respectively (Fig. 2.a) (the "half-time tracking" advocated for in Hewson and Titley (2010)). We use





a simple first order scheme where each cell center in basin A is advected on a sphere by its wind to

$$\phi(t_0 + \frac{\Delta t}{2}) = \phi(t_0) + \frac{v(t_0)}{R_E}\frac{\Delta t}{2} \tag{1}$$

$$\lambda(t_0 + \frac{\Delta t}{2}) = \lambda(t_0) + \frac{u(t_0)}{R_E \cos(\phi(t_0))}\frac{\Delta t}{2} \tag{2}$$

for latitude/longitude $\phi/\lambda$, zonal (u) and meridional (v) velocities, and earth radius $R_E$. If the resulting $|\phi(t_0 + \frac{\Delta t}{2})| > 90°$, the

trajectory has crossed a pole, and we adjust: $\phi(t_0 + \frac{\Delta t}{2}) = 180° \cdot \phi(t_0)/|\phi(t_0)| - \phi(t_0 + \frac{\Delta t}{2}); \lambda(t_0 + \frac{\Delta t}{2}) += 180°$, valid for both poles. The advected location tags the closest cell as within the advected feature. B is advected back in time, and overlap occurs in the commonly tagged cells. For vertical overlap at the common time, the extreme potential temperatures of A and B are held fixed, as if adiabatic. The intersection of the potential temperature ranges is the overlap (Fig. 2.b).

Concisely, Equation 3 defines the similarity between basins A and B at the next time:

$$S(A, B) = \frac{A_{\Delta t/2} \cap_{\phi,\lambda} B_{-\Delta t/2}}{\max(A_{\text{area}}, B_{\text{area}})} + \frac{A \cap_\Theta B}{\max(A_{\Delta\Theta}, B_{\Delta\Theta})} \tag{3}$$

where $\Delta\Theta$ is the maximum amplitude as the sum of intersections in the horizontal and vertical, respectively normalized by the maximum possible overlap so that large areas and amplitudes are most similar to other large areas and amplitudes. Similarity is only considered for basins with horizontal overlap. Note that this also mimics how we create correspondences subjectively, in that we look for features of expected intensities in expected locations. Generically, these correspondences

define candidate connections between overlapping basins and form a directed graph. Note that, technically, the $t_0$, $t_0 + \Delta t$, and common correspondence grids could all be different. Such an option has not been explored but could be beneficial for time-varying or sharply non-uniform meshes.

The type of correspondence is then classified. While TPVs can form, decay, split, merge, and persist over time, we assume that a split or merger event is characterized by one primary and then secondary branches. We categorize the primary and sec-

ondary branches as major and minor correspondences, respectively. A major correspondence is a 1-1 correspondence between A and B where both (1) B is the most similar to A of all the basins at time $t_0 + \Delta t$ and (2) A is the most similar to B of all the basins at time $t_0$. All other correspondences are minor.

Errors in correspondences depend on the imposed similarity function. A basin with small area or amplitude may have less chance of subsequent overlap. When there are multiple TPVs with some overlap, similarity determines major correspondences

quantitatively. While we developed various alternate cost functions to define similarity based on persistence of aspects of a basin's shape over time, each basin metric has weaknesses leading to non-robust associations. For example, radius changes occur when TPVs split, merge, or the basin foothills change. Intensity and amplitude can jump from data assimilation increments or strong interactions with surface lows and terrain. Practically, the inclusion of various metrics is crucial, but relative weighting requires tuning with some level of subjectivity. The (physically motivated) overlap similarity function is at least as

robust for several case studies during development.

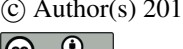



### 2.2.5 Time tracking

To generate tracks for individual TPVs as material eddies with histories of genesis, maintenance, and lysis, major correspondences are connected over time. A track begins when a basin has no major correspondence to a basin at the previous time and ends when a basin has no major correspondence to a basin at the next time. For example, a TPV breaking off from a parent

would be tracked from the timestep after the break through the chain of major correspondences. This approach avoids duplicating the entire history of a track when a TPV splits, with such repeat counting yielding inconsistencies with the actual flow state that can potentially impact subsequent composites and climatologies. To determine if a correspondence at a given time starts a track, we check whether the basin is in a major correspondence at the previous time. Without further information, all major correspondences from the initial time start tracks and to the final time end tracks. The tracks resulting from connecting

major correspondences can then be further filtered as "polar" by restricting to tracks poleward of a jet stream, latitude, or other criteria depending on the application.

Here, only the major correspondences are used, in forming tracks. However, the major and minor correspondences together describe a tree structure of potential overlaps of vortex air masses. For example, this tree has been used to identify all TPVs at future times that could have overlapped with specified TPVs at given times.

### 15 2.2.6 Parameter settings

Default settings are a 300 km filtering disk for regional extrema and $5^{th}$ percentile of the amplitudes of the basin's boundary cells with respect to the core for restriction. Fundamentally, these are imposed filters, with sensitivities that can propagate through the algorithm. Increasing the radius for regional extrema generates larger basins and fewer objects. Increasing the restriction percentile will generate larger basins. Larger basins have more chance for some correspondence overlap and so

may be preferentially favored to last. It is not clear how "optimal" settings would be defined or justified. The default settings best match manual tracks in a small set of case studies. User-defined parameters are exposed in one settings file, and it is straightforward to vary them towards assessing sensitivities within a fixed tracking approach.

### 2.2.7 Computational cost and acceleration

For $0.5° × 0.5°$ ERA-Interim data from 01 January 2010 00 UTC through 30 June 2015 18 UTC every six hours, TPVTrack

run serially takes 25.44 wall-clock hours per tracked year. Of the total run time, pre-processing, segmentation, metrics, correspondence, and tracks take 1.9%, 78.3%, 1.4%, 7.4%, 11.0%, respectively, including input/output. Timings are from an HP Z600 Workstation (eight Intel Xeon Processor E5620s at 2.40 GHz with 11 GB of Cache memory and 16 GB of RAM) with Linux version 2.6.32-504.1.3.el6.x86_64 and Python 2.7.12 from Anaconda.

To reduce wall-clock time, a number of optimizations are possible. A more efficient implementation of a discrete watershed

transform leveraging sorting (Roerdink and Meijster, 2000) could be used for segmentation. While the modules are sequential, each has an embarrassingly parallel decomposition as written (with modifications to file input/output). Within the current version, pre-processing the input data is independent over time; segmentation is independent over time; metrics is independent





over time; correspondence is independent over basin and neighboring times; tracks is independent over track. Each process can write separate, contiguous output files that are then concatenated into one file to maintain consistency with the serial version. Parallel versions of the segmentation and tracks modules have been implemented using Message Passing Interface in Python (Dalcín et al., 2008) with near-linear speedup for each component and identical results to the serial version (for 4 processes).

The only communication between processes is global barriers between components since the modules are sequential.

## 3 Evaluation of TPVTrack

Does TPVTrack accurately represent and track TPVs? Ultimately, any validation (Roache, 1998) depends on a reference truth. We explore this question through both simplified component test cases with reference solutions (Sec. 3.1) and historical cases with more realistic conditions but less clear expectations (Sec. 3.2). A complete discussion, if possible, is beyond the scope

of this paper. Rather, we intend the following to highlight characteristics of the approach and direct future sensitivity analyses and intercomparisons. Users focusing on case studies are encouraged to test robustness of individual tracks, while systematic artifacts may be more relevant for climatologies.

### 3.1 Idealized test cases

Spatial segmentation and time correspondence are central to all approaches. We construct a simplified test case for each

– Sect. 3.1.1 and Sect. 3.1.2, respectively – to better isolate and assess specific differences. TPVTrack is compared to our implementations of Hakim and Canavan (2005) and Kew et al. (2010) as reference point and areal methods, respectively termed H05 and K10.

#### 3.1.1 Spatial segmentation

For segmentation, consider a surface $\theta$ (Fig. 3.a) and vertical relative vorticity $\zeta$ motivated by the Sander's analytic model

(Bluestein, 1992) with

$$\theta(x,y) = \sum_i A_i \cos\frac{2\pi}{L_i}(x+\lambda_i)\cos\frac{2\pi}{L_i}y \tag{4}$$

$$\text{sign}(\zeta) = \text{sign}(\nabla^2\theta) \tag{5}$$

where A=(10,1,0.1) K, L=(1000,100,10) km, and $\lambda = L/4$ mimic various scales of undulations, as for the tropopause (Shapiro et al., 1987). Results are depicted in Fig. 3 on a grid with 50 km spacing.

While the 8 main regional lows are represented in all approaches, the shapes of the objects reflect differences in the algorithms. For H05, the gradient radially outward from the extremum first changing sign defines the bounding contour. Local noise in the surface is amplified by gradients and can drastically shrink objects (e.g., objects near (x,y)=(300,2000) versus (1400,1000) in Fig. 3.b), which may be mitigated with suitable prior filtering. For K10, the maximum area of each object is prescribed ($1.7 \times 10^5$ km$^2$), with smaller objects resulting from overlapping areas (e.g., the two y=100 objects in Fig. 3.c).





While the low and high watershed basins are more robust to local undulations (Fig. 3.d,e) because of the regional extrema filter, the assignment of cells to objects is entirely dependent on the sign of the cell's vertical vorticity. Centered finite differences on the raw field amplify small scales not coherent with the expected larger scale features (Fig. 3.f). Reducing the small-scale noise in the vorticity (Fig. 3.g) is more consistent with the larger scale perspective (Fig. 3.h).

There are also differences in the objects along x=0. For H05, the boundary values are not regional minima given the 650 km search radius. For K10, the 0.5 K contour interval we used for growing each object in turn filters the boundary local minima as parts of existing, more intense objects. For the watershed markers, the 300 km search radius yields boundary objects when aligned with the vorticity. In TPVTrack, this filtering radius is user-defined.

### 3.1.2    Time correspondence

Consider individual disks of radius r and potential temperature range $[\theta_{min}, \theta_{max}]$ advecting about a domain with

$$x(t+\Delta t) = x(t) + u(t) \cdot \Delta t; \quad y(t+\Delta t) = y(t) + v(t) \cdot \Delta t \tag{6}$$

$$u(t+\Delta t) = u(t) + \sigma_u; \quad v(t+\Delta t) = v(t) + \sigma_v \tag{7}$$

$$\theta_{max}(t+\Delta t) = \theta_{max}(t) + \sigma_\theta; \quad \theta_{min}(t+\Delta t) = \theta_{min}(t) + \sigma_\theta; \tag{8}$$

$$r(t+\Delta t) = r(t) + \Delta(\theta_{max} - \theta_{min}) \cdot 10 \tag{9}$$

for position $(x,y)$, velocity $(u,v)$, and additional "noise" increments $\sigma_*$. The increments $\sigma_*$ are a simple way to introduce additional time evolution and are generated independently and stochastically, with the random number generator seeded with 0 for reproducibility. During evolution, we bound $r \geq 10$ km and $\theta_{\max} \geq \theta_{\min} + 1$ K. By construction, a given disk should always and only correspond to itself over time.

For the control case, we initialize 50 objects with 20 km uniform x-y spacing. The initial velocities are sampled from uniform
distributions $u_0, v_0 \in [-5,5]$ m s$^{-1}$, core potential temperatures from a normal distribution $\theta \in N(275,15)$ K, intensities from $\theta_{\max} - \theta_{\min} \in |N(0,15)|$ K, and radii $r \in N(800,100)$ km, motivated by Fig. 6 in Hakim and Canavan (2005). The noise terms sample centered uniform distributions of $\sigma_u, \sigma_v \in [-5,5]$ m s$^{-1}$ and $\sigma_\theta \in [-5,5]$ K. Evolving each cell independently for twelve 6 h timesteps, we can consider the correspondences determined by each approach (Fig. 4). H05 and K10 create numerous false correspondences between nearby objects but miss no correspondences. The stricter 1-1 definition for major
correspondences in TPVTrack greatly reduces the rate of false correspondences but yields several false negatives. Especially since the parameters for the control setup are somewhat arbitrary, several perturbations from the control have been tested. Changes to the initial separation, timestep, velocity amplitudes, and noise amplitudes (not shown) also yield results consistent with the following.

The simple test case illustrates characteristics of each approach. H05 creates correspondences whenever candidates fall
within a distance cutoff. K10 creates correspondences when sufficient horizontal overlap occurs under advection. Correspondences from both are less robust for multiple close objects than TPVTrack, which incorporates horizontal overlap, vertical overlap, and 1-1 major correspondences. Rather than subjectively value one approach as absolutely better than another, these



could be considered complementary tools. Overlap is useful for identifying possibly associated features through time. Further major correspondences use additional information to provide more targeting for following a distinct, individual feature.

## 3.2 Historical test cases

The purposefully simplified segmentation and correspondence test cases permit precise reference solutions. Further evaluation using more realistic historical data centers on two questions: are TPVs tracked by TPVTrack? Are tracks TPVs? Tracks were constructed using the default settings with $0.5° \times 0.5°$ ERA-Interim 2 PVU dynamic tropopause data for 1979 to 2015 every six hours. TPVs are defined as tracks with a core at genesis north of $60°$ N lasting at least 2 days.

### 3.2.1 Are long-lived TPVs tracked?

A track in 2006 is the second longest in this dataset, from 06 July 2006 to 30 September 2006. It is of particular interest because of its long lifetime and associations with Arctic sea ice loss. Tracking manually, we can follow this TPV forming from filamentation and splitting on 03 July over western Siberia to dissipating on 30 September over eastern Siberia after traversing much of the central and eastern Arctic. Using several less robust segmentation and correspondence settings, the TPV's track stops on 29 July, 03 September, 09 September, or 17 September instead. Figure 5 depicts the states driving these sensitivities. Generally, given multiple TPVs with overlap, major correspondences are discriminated quantitatively by similarity scores.

Ullrich and Zarzycki (2017) argue that a closed contour criterion is appealing since it is less sensitive to grid resolution relative to using only nearest neighbors. Here, we quantify some of TPVTrack's spatial sensitivities by segmenting an existing ensemble of full-physics, limited area simulations with model grid spacing from 12 to 120 km. We use WRF-ARW version 3.8.1 (WRF Users, 2016) on a roughly 6000 km by 6000 km domain centered on the north pole with 41 vertical levels up to 10 hPa with a time step of $\Delta t = 5\Delta x$ (e.g., 600 s for 120 km). Initial conditions on 15 August 2006 00 UTC are from GFS analysis with lateral boundary conditions updated every 3 h from alternating GFS 6 h analyses and 3 h forecasts. Physics parameterizations include Morrison 2-moment microphysics (Morrison et al., 2009), Rapid Radiative Transfer Model longwave radiation (RRTM-LW; Mlawer et al. (1997)), RRTM for General circulation models shortwave radiation (RRTMG-SW; Iacono et al. (2008)), Unified Noah land surface model (Tewari et al., 2004), and Yonsei University planetary boundary layer (Hong et al., 2006). The tropopause is diagnosed using NCL functions wrf_pvo to calculate potential vorticity and wrf_user_intrp3 to interpolate to 2 PVU, similar to values found by trapping 2 PVU by searching down from the model top for these grid scales (not shown).

While the number of local extrema increases for the higher-resolution grids within the first day as the model spins up from the external initial conditions, the number of regional extrema remains nearly constant. Comparing the segmentations for the 12, 45, and 90 km grids after 3 days, there are combined effects of dynamical model and segmentation filters (Fig. 6). The higher resolution grid has finer scale structures (Fig. 6.a,b,c) that are preserved in the segmented objects (Fig. 6.e,h,i). A larger radius for regional extrema reduces the number of basins and generally increases their intensities (Fig. 6.e,g). Using the last closed contour over-restricts most of the basins (Fig. 6.d,e,f). Note that over-restriction can be basin-dependent, with the low near (1000, 3200) similar across all panels. Focusing on the long-lived 2006 TPV near (3000, 3000), the segmented basin is



larger for larger cutoff radius, boundary contour percentile, or grid spacing (Fig. 7). The r1000 0% basin at 12 km is spuriously small. The large imposed filter generates few, large regional watershed basins. The minimum potential temperature along the resulting extended boundary is near that of the core, greatly reducing the restricted extent. Vortex cores may also be near the basin's boundary dynamically, like in conditions of strong deformation. In general, the shape of the distribution of potential

temperature on the boundary of the watershed basin determines the sensitivity to the restriction percentile threshold; a uniform distribution with small range would exhibit little sensitivity.

The longest anticyclone track starts on 28 April 1979 and lasts 25 days. Synoptically, a filament from an amplifying Rossby wave break over the northeastern Pacific Ocean develops into a closed feature, couples with numerous cyclones over its entry into and traversal of the Arctic, and dissipates into the polar jet stream east of Greenland. TPVTrack's track exactly matches

our manual track.

### 3.2.2  Bulk evaluation of tracks

Case studies are central to the iterative development of TPVTrack. A common sensitivity, as in the summer 2006 example, occurs when TPVs highly deform or intensify while in a neighborhood with multiple candidate overlaps. Beyond case studies and estimating skill through verification against a limited set of subjective and questionably reproducible manual tracks, bulk

statistics can be used to evaluate methods (Lakshmanan and Smith, 2010). What are the characteristics of tracked TPVs? Are these consistent with expectations? Note that a TPV climatology is beyond the scope of this paper.

While a fuller understanding of the life cycles of TPVs is a matter of ongoing research, Hakim and Canavan (2005) shows TPVs to be generally smaller in radius at genesis or lysis in comparison to the rest of their lifetimes. Both cyclonic and anticyclonic tracked TPVs reach their minimum radius at the beginning or ends of tracks in the majority of cases (Fig. 8).

The tail-dense distributions are also similar for time of minimum amplitude and circulation (not shown). Tracks with larger maximum radius have larger maximum circulation, amplitude, and lifetime ($p < 0.01$).

By construction, tracked TPVs are regional lows in potential temperature with cyclonic vorticity or highs in potential temperature with anticyclonic vorticity. To evaluate TPVs spatially, the Okubo-Weiss parameter (Okubo, 1970; Weiss, 1991) W is:

$$W = s_n^2 + s_s^2 - \zeta^2 \tag{10}$$

where $s_n = \frac{\partial u}{\partial x} - \frac{\partial v}{\partial y}$, $s_s = \frac{\partial v}{\partial x} + \frac{\partial u}{\partial y}$, and $\zeta = \frac{\partial v}{\partial x} - \frac{\partial u}{\partial y}$ are normal strain, shear strain, and relative vorticity, respectively. We include additional metric terms for latitude/longitude grids treating convergence near the poles (Saucier, 1955). If relative vorticity dominates strain, $W < 0$. Maps of W tend to exhibit three regions: rotational vortex cores characterized by negative W, strain regions surrounding vortices characterized by positive W, and background with small absolute W (Provenzale, 1999).

For a TPV-relative reference frame, we center a cardinally-oriented computational grid with uniform 30 km spacing about each TPV core. Values within tracked basins are nearest-neighbor interpolated from the original grid onto the reference grid through a stereographic projection centered on the TPV core. Given the expense of remapping following each TPV, we consider one year (Fig. 9.a,b). Cyclones are frequently rotational about the core, while for anticyclones the region of concentrated rotation

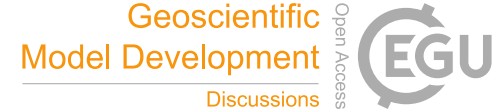



is less frequent, broader, and offset northwards of the core. Considering tracks with at least a 10 days lifetime yields a more rotationally dominated structure about the core (not shown). Geographically, (Fig. 9.c,d), tracked cyclones tend to have negative W except near orographic features, while tracked anticyclones appear to have higher strain at lower latitudes where shear interactions with jet streams would occur.

Thus, in bulk, TPVs from TPVTrack are consistent with eddies with rotational cores exhibiting reasonable genesis and lysis and consistency between metrics. The symmetric, core-relative vortex structure is less frequent for features that are shorter-lived, over mountainous terrain for cyclones, or interacting with lower latitudes for anticyclones. However, it is unclear how discriminating these bulk results are. An intercomparison of trackers would further inform relative algorithm evaluation.

### 3.2.3   Accommodations for model data

Several practical issues have been noted through application of TPVTrack to model data, largely related to the diagnosis of the tropopause and scales of interest. Treatment of missing data via iterative flood-fill (Sect. 2.2.1) is included in the code release. Especially at finer grid scales, relative vorticity may need to be filtered, similar to the "noise" issues seen in the idealized segmentation test case when assigning cells to either highs or lows. In computing metrics, raw or filtered fields could be used, in a trade-off of consistency with the full flow field versus scales of interest.

While a dataset may define its own diagnosis of the dynamic tropopause, conventionally, the dynamic tropopause is found by searching down from some upper level or model top for a potential vorticity threshold within each column independently, possibly with thickness criteria (e.g., Zängl and Hoinka (2001)). However, vertical mixing, folding, and convective-scale vorticity can generate ambiguity with multiple candidates for the tropopause within a column, occurring globally (Añel et al., 2008). Beyond thickness conditions, we have found the tropopause as a surface to be identified more robustly by including consistency

with information outside the column. To identify the tropopause as a layer between the respectively connected high potential vorticity stratosphere and low potential vorticity troposphere, flood-fill segmentations of the stratosphere and troposphere can be performed. For example, starting with initial seed cells near the surface with low potential vorticity, all connected values below a chosen potential vorticity threshold can be found by iterating through valid nearest neighbors. This identification of the connected troposphere yields the tropopause as the top of the tropospheric volume. Note that the top of the troposphere

is above the bottom of the stratosphere, and the bottom of the stratosphere may reach the surface, especially within potential vorticity towers. If model data is archived with sparse vertical levels, this diagnosis may have larger uncertainties. Also note that the chosen PVU level can impact TPV metrics quantitatively (e.g., Fig. 4 in Cavallo and Hakim (2009)).

Typical model output frequencies range from hourly to daily. The simple advection, constant amplitude overlap similarity is less appropriate with decreasing time resolution. Tracking on under-resolved fields may increase systematic misidentifications

of TPVs.

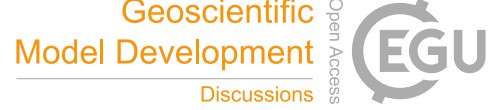



## 4 Conclusions

TPVTrack is a software package for automated tracking of TPVs motivated by limitations of spatial characterization and time tracking within existing approaches. The two primary goals are more robust representations of (1) the spatial structure of features, through restricted regional watershed basins, and (2) mergers and splits in time, through major correspondences from

1-1 similarity overlap. From the simplified test cases, robust segmentation is directly dependent on the vertical vorticity used to associate cells to cyclones and anticyclones, possibly requiring smoothing for noisy fields in pre-processing. Major correspondences reduce the false positive associations but can increase the false negative rate, relative to H05 and K10. Horizontal overlap may benefit from more frequent output or a more sophisticated trajectory model. Including information from more times may make tracking more robust, say with further hypothesis management (e.g., Reid (1979)). Historical test cases reveal

larger uncertainties in tracks and sensitivities to user-defined parameters when multiple TPVs are in a neighborhood.

There has yet to be a methodological study considering whether the approach used to define TPVs impacts our physical understanding. With complementary approaches, several algorithms may be combined constructively to reduce artifacts of any single approach during analysis. In addition to providing another method, TPVTrack's modular design and exposed settings file facilitate intra-algorithm sensitivity studies, e.g., varying the restriction percentile or radius for testing regional extrema. Also,

an alternate method for segmentation (e.g., connected anomalies of minimum intensity or size) could be used as a different model for spatial shape, with no modifications needed for the rest of the algorithm. Perhaps more fundamentally, can a TPV have multiple centers (e.g., Hanley and Caballero (2012))? Users focusing on case studies are encouraged to test robustness of individual tracks, while systematic artifacts may be more relevant for climatologies, including sensitivity to the input data.

The equivalent radius of a TPV's basin measures one aspect of size. Physically, the size of a TPV may refer to a number of

scales, including where fluid is more trapped near the core, the (anti)cyclonic basin about an extremum, minimum distance of the core to a boundary, or the region of flow influenced by the potential vorticity anomaly. Are there clear relationships between these scales? Are different measures more meaningful or useful for different applications? TPVTrack provides one component towards addressing this question.

Towards better understanding of the vertical structure of TPVs, tracks can be constructed on surfaces other than the tropopause.

Associating the separate objects between levels then stitches together the evolution of spatially 3-dimensional objects over time. For a more integrated approach, it may be possible to construct spatially 3-dimensional objects directly by defining appropriate scalings or anomalies for steepest descent in the watershed segmentation and contour restriction. Similar overlap correspondence could then be used for tracking.

Towards tracking features other than TPVs, broadly, TPVTrack identifies coherent objects and creates tracks by overlap

similarity over time, in effect defining the spatial structure and time evolution. Approaches are under development for jet streaks, surface cyclones, cold air outbreaks, and sea ice loss. To allow more flexibility in input data, it may be possible to track on a masked surface where not all values are valid. An application would be to alleviate the uncertainties in estimating sea level pressure over high terrain (e.g., Pauley (1998)) for surface cyclone tracking. Evaluation is needed to inform how the tracked objects map to the physical features for each application, to maximize interpretation of physical results rather than



output from automated algorithms. A general and comprehensive method for evaluation is elusive. Given tracks of multiple features, a central problem is to define and justify when they interact.

*Code availability.* The software can be obtained publicly at https://github.com/nickszap/tpvTrack (doi:10.5281/zenodo.1311001) or from the authors by e-mail.

*Data availability.* The datasets and software are publicly available. ERA-Interim variables can be obtained from the ECMWF data server (https://software.ecmwf.int/wiki/display/WEBAPI/Access+ECMWF+Public+Datasets). The WRF model is available from the WRF Users' page (http://www2.mmm.ucar.edu/wrf/users/), and GFS initial and boundary conditions are available from the National Centers for Environmental Information (https://www.ncdc.noaa.gov/data-access/model-data/model-datasets/global-forcast-system-gfs). In addition, simulation output is available from the corresponding author on request.

*Author contributions.* Both authors have contributed to tracker development, evaluation, and writing of the paper.

*Competing interests.* The authors declare that they have no conflicts of interest.

*Acknowledgements.* The authors thank the members of the University of Oklahoma Arctic and Antarctic Research Group for discussion, especially Christopher Riedel for running the WRF simulations. The authors thank topical editor Juan Antonio Añel for providing constructive comments. This work has been supported by the Office of Naval Research award N00015-1-2220.



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





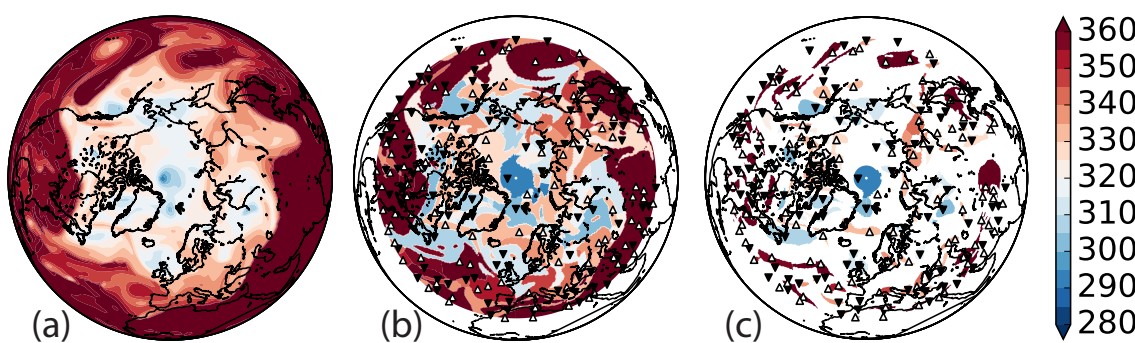

**Figure 1.** (a) Tropopause potential temperature (K; 5 K contour interval) from ERA-Interim on 01 August 2006 00 UTC. (b) Segmentation into low (cyclonic) and high (anticyclonic) regional watershed basins north of $30°$ N. Regional extrema for cyclones and anticyclones are depicted by black downward and white upward triangle markers, respectively. (c) as in (b), with restriction within basins to the $10^{th}$ percentile of boundary amplitudes.



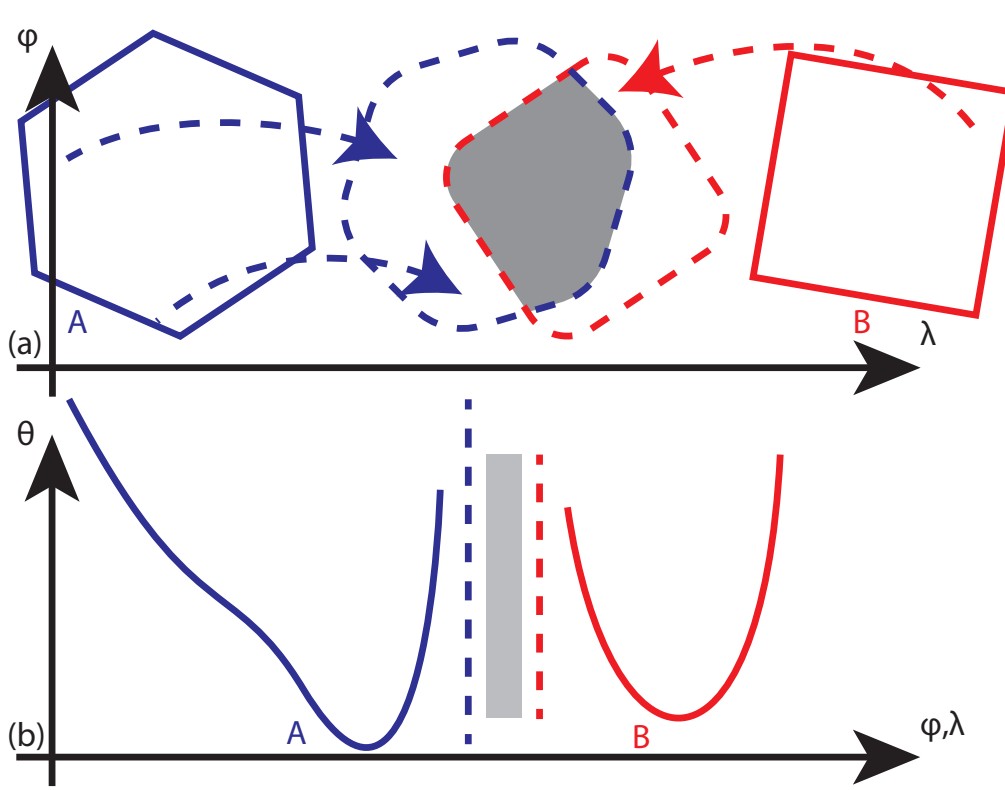

**Figure 2.** Schematic for the two components of time correspondence similarity quantified by overlap (gray) of (a) horizontal cell-to-cell advection and (b) persistent vertical potential temperature ranges for a basin A at $t_0$ (blue) and basin B at $t_0 + \Delta t$ (red) at the intermediate half-time (dashed).





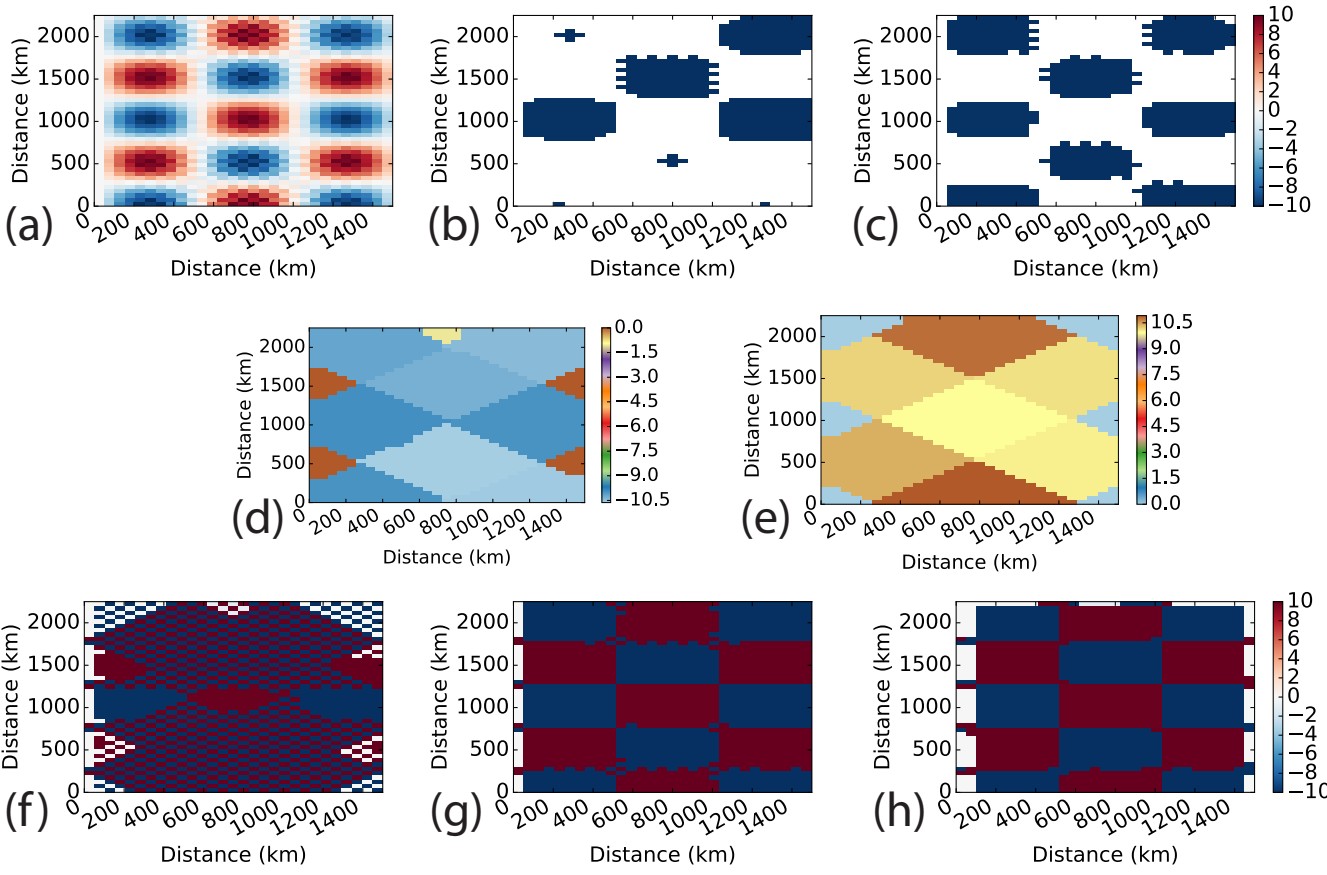

**Figure 3.** Comparison of (b) H05, (c) K10, and watershed (f,g,h) segmentations for an analytic surface with multiple scales (a; Equation 4). For the watershed method, each cell is in both a (d) low and (e) high regional watershed basin. Assignment to one is by the local relative vorticity, calculated by (f) centered finite differences on the raw field (g) a smoother Gaussian Laplacian with standard deviation of 1, and (h) sign of the vorticity is minus the sign of the surface (a). Each object in a segmentation (b-h) is colored by the intensity of its core. Colorbars (K) for (a,b,c,f,g,h) are the same.





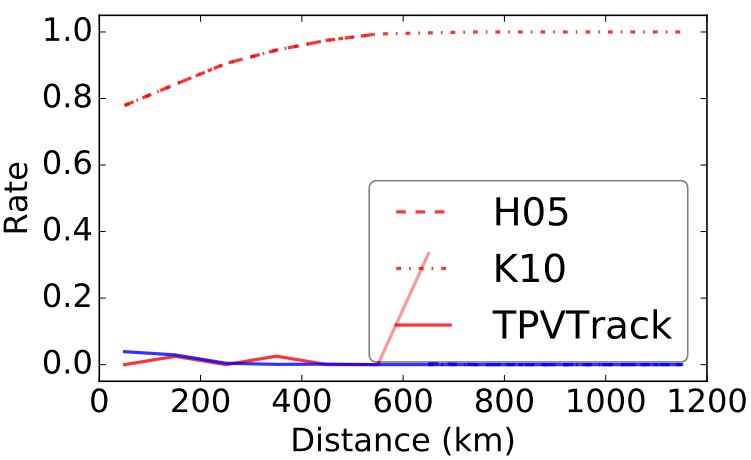

**Figure 4.** False positive (red) and negative (blue) time correspondence rates for control setup of idealized correspondence test case for H05 (dashed), K10 (dash-dotted), and TPVTrack (solid) approaches. Rates are in bins of 100 km core-core distance. For H05 and K10, false negatives are zero and false positives overlap for distances less than 600 km.





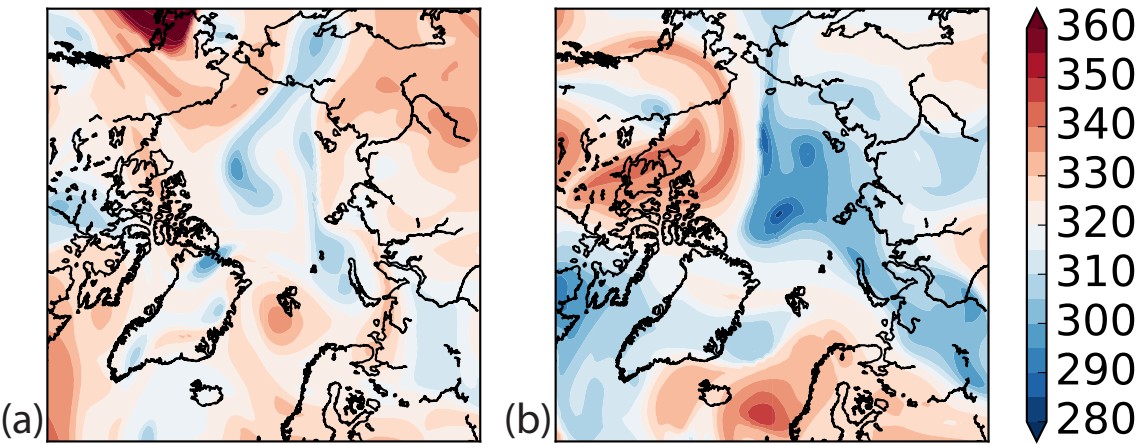

**Figure 5.** Maps of tropopause potential temperature (K; 5 K contour interval) for dates with sensitivities in long-lived Summer 2006 track, (a) 29 July 2006 00 UTC, (b) 17 September 2006 00 UTC




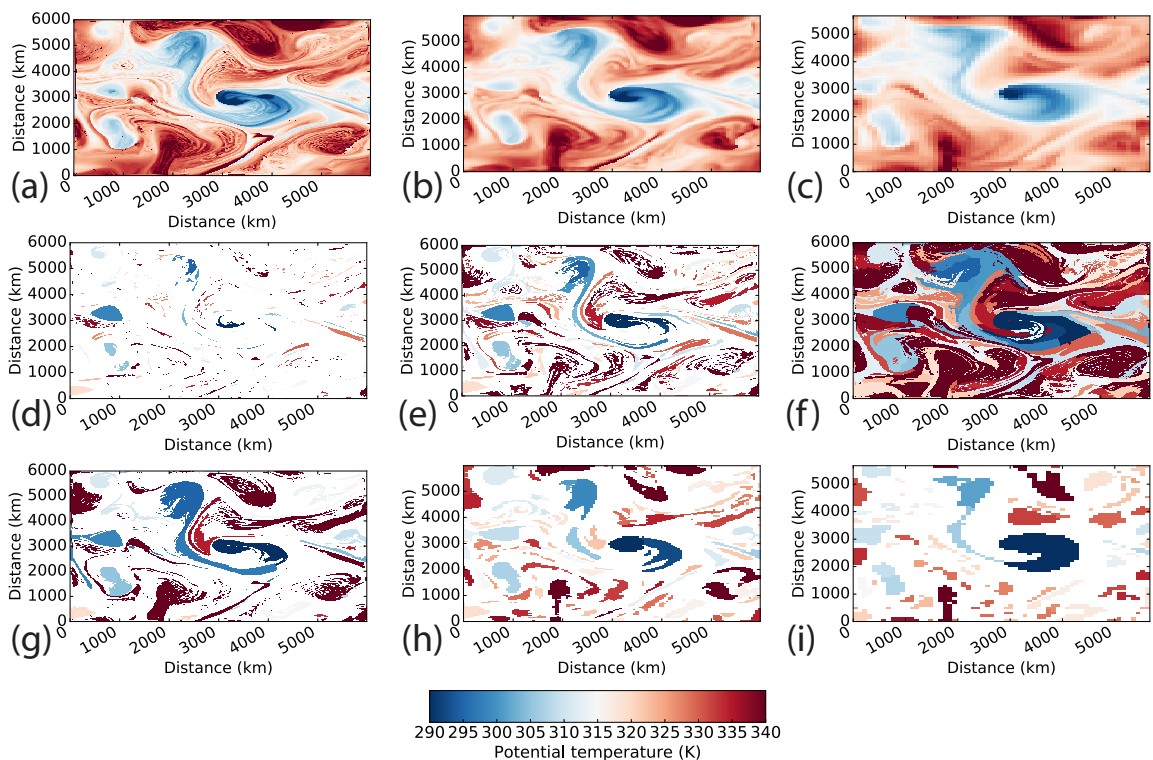

**Figure 6.** Segmentation of (a) 12 km, (b) 45 km, and (c) 90 km WRF tropopause (after 3 simulation days) for (d) 12 km, 0% restriction of boundary amplitudes, and 300 km regional extrema, (e) as in d, but 10% restriction, (f), as in d, but 100% restriction (g) as in e, but 1000 km regional extrema, (h) as in e, but for 45 km simulation, (i) as in e, but for 90 km simulation





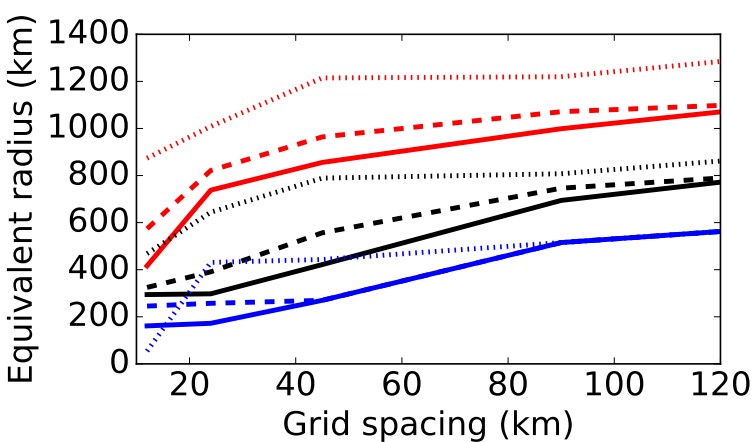

**Figure 7.** Equivalent radius of long-lived 2006 TPV from watershed segmentation of 12 to 120 km WRF simulations (at 3 days) with 300 (solid), 500 (dashed), and 1000 km (dotted) regional extrema and basins restricted to 0% (blue), 10% (black), and 100% (red) boundary amplitudes.





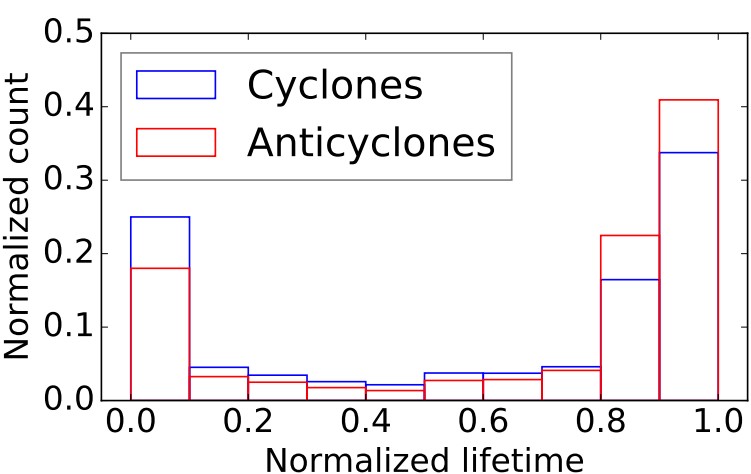

**Figure 8.** Probability of the normalized lifetime (0 at genesis and 1 at lysis within each track) of minimum equivalent radius for cyclonic (blue) and anticyclonic (red) TPVs in 1979 to 2015.





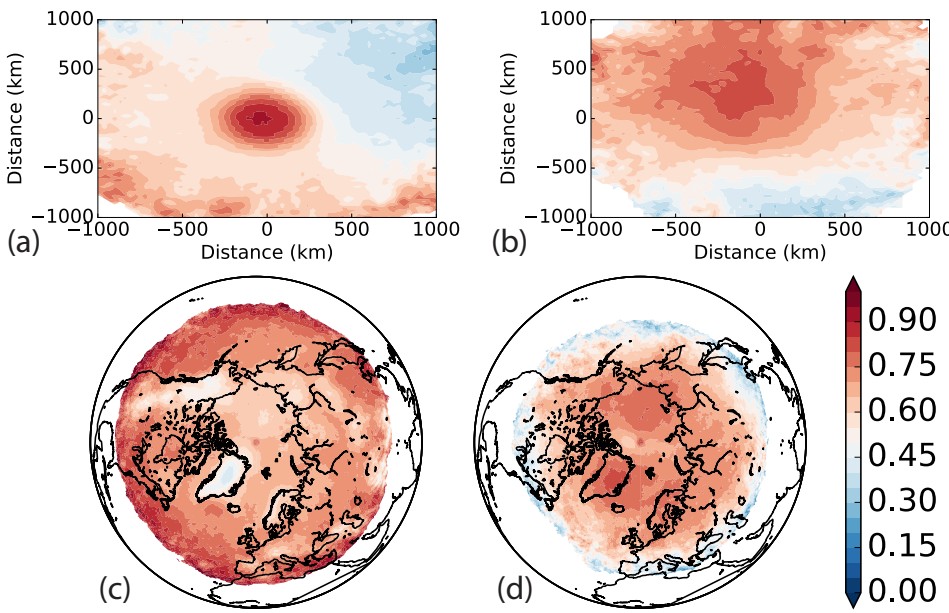

**Figure 9.** Fraction of time with negative Okubo-Weiss given a TPV present, (a) cyclonic TPVs in 2006 in a core-relative reference frame, (b) anticyclonic TPVs in 2006 in a core-relative reference frame, (c) cyclonic TPVs in 1979 to 2015 geographically, and (d) anticyclonic TPVs in 1979 to 2015 geographically. Values with fewer than 5 days of samples (on the domain boundaries) are masked in white. The contour interval is 5%.