# Peer review of "TPVTrack v1.0: A watershed segmentation and overlap correspondence method for tracking tropopause polar vortices"

_Geoscientific Model Development, 2018_

## Referee Comment (RC1) · B. Langenbrunner (Referee) · 23 Sep 2018

Review of "TPVTrack v1.0: A watershed segmentation and overlap correspondence method for tracking tropopause polar vortices," by Nicholas Szapiro and Steven Cavallo

This paper presents a new method and accompanying Python package for tracking tropopause polar vortices/TPVs. Generally, I find the paper very clear and well written, and as a non-expert on TPVs, I was able to follow it quite easily. I also think the authors did a nice job of differentiating their work from that of previous studies.

Because the subject matter is far from my area of expertise, I'm not able to comment on how this method compares to the others discussed in the paper. Instead, I was asked to review the Python-related aspects of this manuscript, and my comments and suggestions below are related to this.

I understand that a large amount of work goes into making a package like this portable, and my experience cloning the GitHub repository and changing parameters to run the "demo" case was fairly easy, though I did spend some time troubleshooting, and I've provided some more text to make proper adjustments below. Overall, I think this paper reads well and that the accompanying GitHub repository, with some minor changes, is suitable for publication in GMD. I don't think the major suggestions ("BROAD/MAJOR COMMENTS") need to be addressed for publication; I intend for them to be constructive comments for the authors to consider down the line.

GENERAL QUESTIONS/COMMENTS

This code repository seems very well written and logically structured, though a general comment is that while looking through the .py scripts, I desired a bit more documentation throughout. For example, a docstring below each function's definition (def . . .), briefly stating what it does (and/or its input/output) would be helpful in troubleshooting and reading the source code. See https://www.python.org/dev/peps/pep-0257/.

I only tried the simplest demo case with the provided ERA-Interim data set, and I haven't experimented with other data or running in parallel, so this review concerns the "unified" branch on GitHub.

BROAD/MAJOR COMMENTS (FOR FUTURE VERSIONS OF CODE)

I don't think this is important for the current manuscript, but I imagine it will eventually be useful to migrate this package to Python 3.x. I know Python 2.7 has a large user base, but using 3.x would likely give it more portability and would make it easier to use with other packages that are being widely adopted in the NetCDF community like

xarray and dask (for parallel computing).

In the same vein, I think it will also be good to migrate the plotting aspects over to cartopy eventually, since basemap has an end-of-life date sometime in 2020 (see the announcement here: https://matplotlib.org/basemap/users/intro.html#cartopy-new-management-and-eol-announcement). Again, not important for the current manuscript, but something to consider for future versions.

tpvTrack works great as a collection of scripts, but it's not precisely at the level or structure of a Python package. I take it that this is not the authors' intention for this initial publication, but if that is an eventual goal, some restructuring would probably need to take place. There's a nice tutorial for this here: https://packaging.python.org/tutorials/packaging-projects/.

The documentation is well done and easy to read, though I can see that it's a work in progress. If the authors hope that this package grows and is adopted by a broader user base, they might consider creating a documentation on Read the Docs using sphinx. This is not necessary for the current version, but again, something to consider for future versions and maintenance.

SUGGESTED MINOR CHANGES TO DOCUMENTATION

A few other things that would help clarify how the package works within the documentation would be:

* A clear list of output from each of the different options at the bottom of driver.py. I used demo() and demo_algo_plots(), but I didn't experiment much with the others. Also, adding some language in the documentation about the plots that are created and the .nc files that are produced would make them more user friendly. I wasn't sure what "corr_debug_X.png" and "seg_X_...png" corresponded to at first.

* A sentence somewhere stating that .pyc files will be created (but ignored in .gitignore) while running driver.py would be useful.

* The documentation only lists the main modules, but I would recommend adding the minor ones, as well, to be comprehensive.

* When running the demo, the "info" variable should also be changed so that global .nc attributes are appropriate, so this ought to be added in section 5.

SUGGESTED CHANGES TO CODE/REPOSITORY

The changes below helped me to successfully run the demo. It's totally up to the authors whether they want to implement all of these changes.

* I cloned and forked the GitHub repository and ran into a few issues with dependent package versions. To make the Anaconda environment easy to reconstruct, I would therefore recommend that the authors provide an environment (.yml) file that details the package versions that are installed. Then, Anaconda users could quickly install a new environment and (ideally) be up and running more quickly. For example, basemap doesn't work when installed using its default channel (there's a matplotlib issue with "set_axis_bgcolor()", which is a deprecated function). Instead, I needed to download it from conda-forge. The authors can create an environment file quickly using this tutorial: https://conda.io/docs/user-guide/tasks/manage-environments.html#sharing-an-environment.

* It might be a good idea to add a test-tpvTrack directory explicitly to the repository, so that users can work with the package out of the box using the default ERA-Interim data set. For example, as a user, I would ideally like to download the package and see a filled test-tpvTrack directory with successful output. The pull request I've created has these additions. This makes the GitHub repo slightly larger ($\sim$100 MB now, given the additional .nc files that driver.py creates), but I think it's still a reasonable size and well within the hosting limits and allows users to see what a "successful" run should look like.

* In changing the file variables for the package to properly run the demo, I noticed the

documentation says to modify things like fDirData in preProcess.py, but I had to modify these variables in my_settings.py. I ended up modifying my_settings.py and would recommend that the authors change section 5 in the documentation to reflect this. I also changed the directory variables (like fDirData) to use relative links rather than full links, which may help users run the demo code with fewer modifications.

* I changed the color map in the segment.py script to use the RdBu_r color bar, so that it looks closer to Fig. 1 of the manuscript.
* * *

---

## Referee Comment (RC2) · Anonymous Referee #2 · 18 Oct 2018

**Reviewer's summary of the manuscript**

In "TPVTrack v1.0: A watershed segmentation and overlap correspondence method for tracking tropopause polar vortices, " Szapiro and Cavallo present a new software framework for detecting and tracking tropopause polar vortices (TPVs). They qualitatively describe TPVs as persistent areas of high potential vorticity (positive and negative) occurring along the tropopause that are associated with the broader polar vortex. They describe two other TPV tracking methods and give overview of the various considerations involved in robustly detecting and tracking features like TPVs.

[Figure]

The authors summarize their choices for tracking methodology as: (1) a watershed segmentation model on potential vorticity (for negative and positive anomalies separately), combined with (2) an advection-overlap method for ascertaining temporal continuity of detected vortices. The method considers TPVs as detected tracks that persist for 2 or more days and that occur poleward of 60 degrees. They describe a number of geometric and dynamic metrics that TPVTrack v1.0 can calculate.

Szapiro and Cavallo apply TPVTrack v1.0 to an idealized spatial field of potential vorticity–with added noise–to compare the method with two other tracking methods and to demonstrate that in principal TPVTrack v1.0 identifies vortices in a way consistent with the authors's descriptions of the tool. They further apply the tool to a specific synoptic case that involves simulations with WRF at multiple resolutions. Finally, they apply the tool to one year's worth of ERA-Interim output and examine vortex-centered composites of TPVs to show that their detected TPVs fit dynamical expectations.

**Summary of Review**

The manuscript presents a well-written and thorough description of TPVTrack v1.0, which appears to be a valuable new tool for researchers desiring to track tropopause polar vortices. The methodology described is sufficiently novel compared to other methods, and most of the methodological choices seem logically sound. With a few notable exceptions (described below), the figures are well-described and support the manuscript text. Overall, the manuscript is nominally worth publication in some form.

However, despite this, I have a few major concerns that in combination may preclude publication in GMD in its current form:

* lack of compelling scientific application of the new tool, * superficial discussion of the reasoning behind some key methodological choices, and * superficial discussion of uncertainties and assumptions involved in the methodological choices. * inadequate discussion about how the results of the method might (or actually do) depend on technical details of the model used (e.g., especially horizontal resolution)

Overall, these concerns combine such that it seems to me that this paper is not appropriate for the aims and scope of GMD. Relatedly, I am not sure it would be of general interest to the GMD readership, since the intended audience appears to be solely dynamicists with interests in tracking TPVs, which I expect is a narrow portion of the readership.

The manuscript comes across to me as a sound technical description of a new software tool. While GMD does support manuscripts that offer technical descriptions, these–as far as I am aware–are technical descriptions of new geophysical models. Instead, this manuscript provides technical description of a tool for model analysis. It is not clear to me that this type of manuscript is appropriate for GMD. It would however be a very obvious candidate for publication in a journal like The Journal of Open Source Software.

Even if GMD does support publication of this category of article, I would still suggest that the manuscript warrants major revisions due to the reasons enumerated above. In its current form, the manuscript lacks a compelling scientific application that would aid readers in seeing value and relevance of this method, and the lack of discussion about methodological choices effectively makes for results that are not repeatable by others. For example, if a reader decided to implement this TPV tracking method and had to make a choice about some specific implementation details ('TPVs are defined as tracks with a core at genesis north of 60N lasting at least 2 days' – what was the basis for the authors deciding on this definition of tracks???), the manuscript does not provide sufficient detail for the reader to come to the same conclusion as the authors about the implementation details given in the manuscript. This also then precludes a reader from debating the authors's conclusion about the choice of these details, since details about the choice are not given. Related to this, some of these implementation choices have some amount of uncertainty associated with them (e.g., why 60N and 2 days versus 61N and 1.5 days?), but the authors do not discuss the implications of these uncertainties. This is critical, since there is a growing recognition in the litera-

ture that such details may have enormous and important uncertainties associated with them (e.g., see two recent papers on the Atmospheric River Tracking Method Intercomparison Project: https://journals.ametsoc.org/doi/abs/10.1175/BAMS-D-18-0200.1 and https://www.geosci-model-dev.net/11/2455/2018/gmd-11-2455-2018.html).

Additional details of these criticisms follow.

**Major issues**

**No Scientific application**

The manuscript is completely focused on a technical description of the TPV tracking tool. Though GMD does support technical descriptions of models, my experience as a reader is that technical papers are much more useful if they also provide a simple scientific use case that illustrates the value of the new method/model. Without this, the scientific value of this particular TPV tracking method is not clear.

On pg 12, line 16, the authors state "that a TPV climatology paper is beyond the scope of this paper." My initial reaction to seeing this is 'that is a shame'; it seems like it would be a very easy target for the authors to ask the simple questions of "what is the climatology of TPVs?", "how does the climatology from our method compare with that from the other methods that exist in the literature?", and "if there is a difference, why might this method be more valid?" The authors even state on page 11 that they ran their algorithm on ERA-interim from 1979 to 2015: so why not show any of these results? In my opinion, arguing that this is beyond the scope of the paper harms rather than helps the paper.

Note that the authors do seem to have an interesting result associated with Figure 8, but they only devote one sentence to that figure. If I understand the result correctly (but see my comment in the 'Minor Issues' section below, noting that I'm not convinced I do), this implies that TPVs grow, reach a maximum size some time in the middle of their life cycle, and then shrink again. This seems interesting and potentially worth digging

in to a bit more. Is this expected? Would other tracking methods show a similar result?

**Lack of Justification for Methodological Choices**

In general, the authors do a good job of describing their reasoning behind key methodological choices: e.g., the watershed basin method is more robust to 'grid scale undulations' in the field. However, there are a four key choices for which inadequate justification is provided:

1. 'TPVs are defined as tracks with a core at genesis north of 60N lasting at least 2 days' (pg 11, line 7) * TPV minimum latitude: 60N * TPV minimum duration: 2 days 2. 'Default settings are a 300 km filtering disk for regional extrema and 5th percentile of the amplitudes of the basin's boundary cells with respect to the core' (pg 8, lines 16-17) * TPV filtering disk radius: 300 km * TPV percentile threshold: 5th percentile

Of these four choices, the TPV minimum latitude and TPV minimum duration are the least justified: as far as I can tell, the authors simply state this choice without qualification. If another researcher implemented this method, the paper provides no information about why the researcher should conclude that 60N and 2 days should be the default values. Also, these choices effectively embed assumptions about the nature of TPVs in them, and these assumptions (and their implications) should be explicitly stated.

The TPV filtering disk radius and TPV percentile threshold do have some explanation provided; the authors appropriately explain that 'Increasing the radius for regional extrema generates larger basins and fewer objects. Increasing the restriction percentile will generate larger basins' (pg 8, lines 18-19). However, the authors also state that 'The default settings best match manual tracks in a small set of case studies.' This seems like a reasonable basis on which to make parameter choices, but what small case studies and which manual tracks? Without this information, a reader has no chance of evaluating whether they agree that the given choices do result in a good match with case studies and that another choice of parameters would be inferior. I assume that the authors are partly forward referencing the result stated on Pg 12, lines

9-10: "TPVTrack's track exactly matches our manual track," and if so, this should be made explicit. However, this is only one track (the authors indicate that more than one track is used for deciding on parameters), and the track is not actually shown: these manual track are critical data on which the authors are making methodological choices and so should be included in the paper (perhaps as supplementary material?).

**No Discussion of Uncertainties Associated with Methodological Choices**

Related to the above, the authors do not adequately discuss the implications of uncertainties associated with these parameter choices. As noted above, the authors do discuss the effects of varying the filtering disk radius and percentile threshold on individual fields. This is a good direction for discussion, and it would be useful if the authors could expand this discussion to (1) include discussion of implications for changing other parameters, and (2) expand the discussion further to include implications for climate-length studies. It would be even more useful if the authors directly showed the effects of these parameter choices on climatological TPV track information.

This is particularly important as there is a growing recognition in the literature that this uncertainty can have major implications for our understanding of weather and climate. For example, the IMILAST project (extratropical cyclone detection intercomparison) shows a ~6x variation in the counts of cyclones across 15 different methods: https://doi.org/10.1175/BAMS-D-11-00154.1. Likewise, ARTMIP (atmospheric river detection intercomparison) a similarly large spread in AR statistics across numerous detection methods (https://journals.ametsoc.org/doi/abs/10.1175/BAMS-D-18-0200.1 and https://www.geosci-model-dev.net/11/2455/2018/gmd-11-2455-2018.html). The ARTMIP project is currently working on experiments to understand whether these different algorithms might produce different climate sensitivities for ARs in climate change experiments. Given this growing understanding in the literature, it is critically important that tracking-method papers such as this explicitly explore uncertainty at the outset. I'm not arguing that the authors should tackle an intercomparison of the scale of IMILAST or ARTMIP, but since, as the authors note on pg 8 lines 21-22, TPVTrack makes it easy

to explore parametric uncertainty, the authors should do just that.

**Minor issues**

pg 1, lines 24-25: "Diagnostic trajectories and prognosed scalar transport further support the advection-dominated dynamics for individual cases (not shown)." <– It is very odd to include a new, not-shown result in the intro: why do this?

pg 2, lines 4–6: This section should reference IMILAST and ARTMIP, which are both very relevant to the discussion

pg 2, lines 18–20: Regarding the first sentence of this paragraph: from where does this qualitative definition originate? If there is a common source (e.g., a textbook), it should be cited. If not, would other polar dynamicists agree on this? Marty Ralph had to convene two AGU townhalls to come to a qualitative, consensus definition of ARs (which is now in the AMS glossary): why would TPVs be different? If this definition is original, I would suggest a rephrasing to make clear that this is a proposed definition: e.g., "We propose a functional, qualitative definition of TPVs: ..."

pg 2, lines 20–21: "...are fundamental to an automated scheme" <–I would argue this is true for any objective, quantitative scheme: whether automated or not.

pg 4, line 6: "through a modular, object-oriented approach is publicly available" <– There seems to be a word missing in this sentence (should it be "*which* is publicly available"?)

pg 8, line 8: 'It is not clear how "optimal" settings would be defined or justified. The default settings best match manual tracks in a small set of case studies.' <– These two sentences seem to contradict each other. The first says we don't know how to define 'optimal', and the second says that we used a small case study to show that our parameter setting results in the best match (which sounds 'optimal' to me).

pg 8, line 30: "...; metrics is independent" ('is' should be 'are')

pg 11, line 25: "similar to values found by trapping 2 PVU by searching down from the model top for these grid scales" <– I have no idea what this means. I would suggest rephrasing somehow.

pg 11, line 30: "(Fig 6.e,h,i)" <–Is the lettering here what was actually intended? I'm having a hard time understanding what the authors are referring to.

pg 12, lines 9–10: "TPVTrack's track exactly matches our manual track." <–What manual track? I see no figure for this.

pg 12, lines 18–19: "Both cyclonic and anticyclonic tracked TPVs reach their minimum radius at the beginning or ends of tracks in the majority of cases (Fig. 8)." I struggled to see how Figure 8 indicates this. It's not that I doubt the result, but rather that the caption for Figure 8 doesn't make sense to me and/or the axis labels are confusing.

pg 13, line 25: "...and the bottom of the stratosphere may reach the surface" <– What!? Perhaps there is a polar atmospheric phenomenon that I've not yet learned about, but I've never heard of the tropopause reaching the surface in any dynamical circumstance. I'm wondering if the wording here conveyed something that the authors didn't intend. Otherwise, if this can actually happen, a reference here should be added, since I expect I wouldn't be the only reader to be surprised to learn this.

Figure 2b: I read the text and caption several times and I still can't figure out what Figure 2b is supposed to convey.

Figure 6: Titles/labels on the subplots would be extremely useful. Given that there are 3 resolutions, my initial inclination was to think that columns correspond to resolution– but this isn't true (d,e,f). Because of this confusion, I found I had to repeatedly keep looking between the figures and the caption to understand what I was looking at. It doesn't help that the captions for the subplots reference other parts of the caption ("(i) as in e, but for..."). I found I spent way more time going back and forth between the caption and figures than I normally do in a paper, which made this quite frustrating–and
I don't think it needs to be.

Figure 8: I think the caption needs to be reworded. I went back and forth between the figure and the text multiple times before I think I understood the figure. If I understand it correctly, it might be more usefully worded as "Probability of TPVs being at their minimum equivalent radius as a function of lifecycle for cyclonic...". Also, for the label of the horizontal axis, I would suggest the word 'lifecycle' rather than 'lifetime', because the term lifetime made me think that the horizontal axis refereed to a measure of the duration of the TPV relative to other TPVs.
* * *

---

## Author Comment (AC1) · 17 Nov 2018

We thank the reviewers for the constructive comments towards improving the work and include the contributions in the Acknowledgments. As requested, we submit the revised files, a point-by-point reply to the comments, and a marked-up manuscript showing the changes using latexdiff. Response to RC1 is included inline below. Response to RC2 is attached as a supplement in pdf format.

Sincerely, Nicholas Szapiro and Steven Cavallo

In response to RC1: We thank Baird Langenbrunner for review and improvement of the software package and documentation. In addition to his modifications incorporated via pull request, we include additional software changes for the unified branch, described below in response to the review point-by-point. The manuscript has been changed to acknowledge the review.

More documentation: Docstrings below function definitions now include a brief description of what the function does and the input arguments if possibly ambiguous.

Code portability: Migration to Python 3, cartopy, and a Python package are welcome suggestions for development and incorporation into future versions.

Clear list of output: An output section has been added to the User's Guide, describing output from demo() and demo_algo_plots(). The other options at the bottom of driver.py were products of older versions. They have been removed, with driver.py correspondingly cleaned.

"Note that .pyc files will be created (but ignored in .gitignore).": The sentence has been added to the User's Guide in the ERA-Interim test case section.

Documentation contents: With more comprehensive documentation a goal of future development (as a Python package), description of the core modules in the manuscript and User's Guide is intended to further orient the reader. We direct users to the (readable) source code if the desired implementation details extend beyond what is described in the documentation. NetCDF files output from preProcess, basinMetrics, and tracks now include units and long_name metadata. The segmentation and correspondence files contain descriptions metadata.

"info" variable: Changing the "info" variable is now a step in the ERA-Interim test case in the User's Guide.

Reconstruct Anaconda environment: A docs/environment.yml is now included and mentioned in the User's Guide.

test-tpvTrack directory: The test-tpvTrack directory with output from the example test case was added with the reviewer's pull request. Files have been updated to match output from the current version.

Changes in my_settings.py: The User's Guide now correctly instructs the user to modify variables in my_settings.py.

Color map: The colormap was changed with the reviewer's pull request.

Please also note the supplement to this comment:
https://www.geosci-model-dev-discuss.net/gmd-2018-180/gmd-2018-180-AC1-supplement.pdf

**Supplement:**

Response to Anonymous Referee #2

We thank the reviewer for improving the clarity, content, and quality of the manuscript. Point-by-point replies are included inline below, with the reviewer's text in black and our responses in blue. Note that Figs. 8 and 11 have been added, so the previous Figs. 8 and 9 are now Figs. 9 and 10, respectively.

**Reviewer's summary of the manuscript**

In "TPVTrack v1.0: A watershed segmentation and overlap correspondence method for tracking tropopause polar vortices, " Szapiro and Cavallo present a new software framework for detecting and tracking tropopause polar vortices (TPVs). They qualitatively describe TPVs as persistent areas of high potential vorticity (positive and negative) occurring along the tropopause that are associated with the broader polar vortex. They describe two other TPV tracking methods and give overview of the various considerations involved in robustly detecting and tracking features like TPVs.

The authors summarize their choices for tracking methodology as: (1) a watershed segmentation model on potential vorticity (for negative and positive anomalies separately), combined with (2) an advection-overlap method for ascertaining temporal continuity of detected vortices. The method considers TPVs as detected tracks that persist for 2 or more days and that occur poleward of 60 degrees. They describe a number of geometric and dynamic metrics that TPVTrack v1.0 can calculate.

Szapiro and Cavallo apply TPVTrack v1.0 to an idealized spatial field of potential vorticity–with added noise–to compare the method with two other tracking methods and to demonstrate that in principal TPVTrack v1.0 identifies vortices in a way consistent with the authors's descriptions of the tool. They further apply the tool to a specific synoptic case that involves simulations with WRF at multiple resolutions. Finally, they apply the tool to one year's worth of ERA-Interim output and examine vortex-centered composites of TPVs to show that their detected TPVs fit dynamical expectations.

**Summary of Review**

The manuscript presents a well-written and thorough description of TPVTrack v1.0, which appears to be a valuable new tool for researchers desiring to track tropopause polar vortices. The methodology described is sufficiently novel compared to other methods, and most of the methodological choices seem logically sound. With a few notable exceptions (described below), the figures are well-described and support the manuscript text. Overall, the manuscript is nominally worth publication in some form.

However, despite this, I have a few major concerns that in combination may preclude publication in GMD in its current form:

* lack of compelling scientific application of the new tool, * superficial discussion of the reasoning behind some key methodological choices, and * superficial discussion of uncertainties and assumptions involved in the methodological choices. * inadequate discussion about how the results of the method might (or actually do) depend on technical details of the model used (e.g., especially horizontal resolution)

We respond to these 4 major concerns in the corresponding, more-detailed sections below.

Overall, these concerns combine such that it seems to me that this paper is not appropriate for the aims and scope of GMD. Relatedly, I am not sure it would be of general interest to the GMD readership, since the intended audience appears to be solely dynamicists with interests in tracking TPVs, which I expect is a narrow portion of the readership.

We agree that dynamicists with interests in tracking TPVs are an intended audience of this manuscript. We disagree that they are the sole audience.

TPVs are a class of upper level potential vorticity anomalies usually present in the Arctic. Upper level and surface PV anomalies are classically important in mid-latitude synoptic meteorology. With further connections to and from TPVs throughout the Earth system possible (as outlined in the paper's introduction), additional recent work connects TPVs with cold air outbreaks (Biernat 2017). Connections with atmospheric rivers may be possible as well but have not yet been pursued to our knowledge.

We expect that furthering our understanding of these connections will be fruitful. Publication of TPVTrack in GMD facilitates these goals. Moreover, TPVTrack can be adapted to track other features as mentioned in the Conclusions (with modification done in Biernat 2017 for synoptic cold pools, for example), potentially of additional interest to the GMD readership.

The manuscript comes across to me as a sound technical description of a new software tool. While GMD does support manuscripts that offer technical descriptions, these–as far as I am aware–are technical descriptions of new geophysical models. Instead, this manuscript provides technical description of a tool for model analysis. It is not clear to me that this type of manuscript is appropriate for GMD. It would however be a very obvious candidate for publication in a journal like The Journal of Open Source Software.

As the reviewer does not further define geophysical models and model analysis or differentiate between them, we consider two definitions. Under the current AMS Glossary's definition

(http://glossary.ametsoc.org/wiki/Model), TPVTrack fits in the "heuristic method" category as a pattern recognition technique as an application/adaptation of image processing strategies. More broadly, if a geophysical model is a scientific model that takes geophysical input and generates geophysical output, where "[s]cientific modelling is a scientific activity, the aim of which is to make a particular part or feature of the world easier to understand, define, quantify, visualize, or simulate by referencing it to existing and usually commonly accepted knowledge" (https://en.wikipedia.org/wiki/Scientific_modelling), TPVTrack qualifies as well. In short, TPVTrack takes input of atmospheric fields, defines physical features termed tropopause polar vortices through rule-based patterns, and outputs a history of features to be post-processed by the user. It is a component towards furthering our understanding of TPVs and their interactions with the Earth system. That is, TPVTrack is a geophysical model.

However, if it is still argued that TPVTrack does not qualify as a geophysical model but rather "a tool for model analysis" instead, we believe that this manuscript would still be suitable for publication under GMD's Method for assessment of models manuscript type as both a software tool and discussion of a novel method for data analysis (https://www.geoscientific-model-development.net/about/manuscript_types.html#item3). In this alternate view, TPVTrack is a diagnostic tool to add TPV-related state variables to the model outputs, where the output data can then also be used independently.

While we are not familiar with the distribution of interests of GMD readership, "CycloTrack" (as a model description) and "TempestExtremes" (as a method for assessment of models) are similar works focusing on surface cyclones that have been published in GMD.

Even if GMD does support publication of this category of article, I would still suggest that the manuscript warrants major revisions due to the reasons enumerated above. In its current form, the manuscript lacks a compelling scientific application that would aid readers in seeing value and relevance of this method, and the lack of discussion about methodological choices effectively makes for results that are not repeatable by others. For example, if a reader decided to implement this TPV tracking method and had to make a choice about some specific implementation details ('TPVs are defined as tracks with a core at genesis north of 60N lasting at least 2 days' – what was the basis for the authors deciding on this definition of tracks???), the manuscript does not provide sufficient detail for the reader to come to the same conclusion as the authors about the implementation details given in the manuscript.

We respond in the major issues section below. Note that, using the included TPVTrack software and available input data, all methods and results are reproducible. We invite competing or

alternate conclusions regarding the formulation, implementation, and internal settings of TPVTrack.

This also then precludes a reader from debating the authors's conclusion about the choice of these details, since details about the choice are not given. Related to this, some of these implementation choices have some amount of uncertainty associated with them (e.g., why 60N and 2 days versus 61N and 1.5 days?), but the authors do not discuss the implications of

these uncertainties. This is critical, since there is a growing recognition in the literature that such details may have enormous and important uncertainties associated with them (e.g., see two recent papers on the Atmospheric River Tracking Method Intercomparison Project: https://journals.ametsoc.org/doi/abs/10.1175/BAMS-D-18-0200.1 and https://www.geosci-model-dev.net/11/2455/2018/gmd-11-2455-2018.html).

We respond in the major issues section below.

Additional details of these criticisms follow.

**Major issues**

**No Scientific application**

The manuscript is completely focused on a technical description of the TPV tracking tool. Though GMD does support technical descriptions of models, my experience as a reader is that technical papers are much more useful if they also provide a simple scientific use case that illustrates the value of the new method/model. Without this, the scientific value of this particular TPV tracking method is not clear.

An additional use case has been added to the introduction. In the Conclusions, scientific value is now better communicated by presenting (1) selected results from the default settings and (2) concerns over use of a single approach.

This manuscript is focused technically in order to document the new method. The manuscript also stresses that a unique method may not be the best strategy for accurate representation of TPVs in all respects and complementary approaches may be fruitful. Originally, TPVTrack was initiated to address limitations in H05 in terms of robustness of spatial shape to small scale noise significant to local gradients (which is particularly important for higher-resolution datasets) and 1-1 temporal history (which is particularly important for TPVs, which are UTLS

features that can "live" for months). These problems also plague other existing approaches. Exploration of the physical basis versus artifact character of existing approaches led to TPVTrack, and we hope the future holds further progress as well.

On pg 12, line 16, the authors state "that a TPV climatology paper is beyond the scope of this paper." My initial reaction to seeing this is 'that is a shame'; it seems like it would be a very easy target for the authors to ask the simple questions of "what is the climatology of TPVs?", "how does the climatology from our method compare with that from the other methods that exist in the literature?", and "if there is a difference, why might this method be more valid?" The authors even state on page 11 that they ran their algorithm on ERA-interim from 1979 to 2015: so why not show any of these results? In my opinion, arguing that this is beyond the scope of the paper harms rather than helps the paper.

We agree that an updated TPV climatology is worthwhile and have added "Note that a TPV climatology is beyond the scope of this paper and the focus of a separate work." This publication is one of a series of papers from members of the Arctic and Antarctic Research Group at the University of Oklahoma. Another graduate student is leading paper(s) on such a climatology, with discussion of input reanalysis, seasonality, associations with teleconnections, long-term trends, significance of differences, and sensitivity to tracking method. It is not clear to us that separating the climatology as a separate work harms this paper, and the separation benefits the climatology through fuller treatment.

Note that Sect. 3.2 Historical test cases is based on tracks using ERA-Interim input data from 1979 to 2015. The vortex-centered composites consider only one year of the historical period to reduce the cost of re-centering about each TPV.

Note that the authors do seem to have an interesting result associated with Figure 8, but they only devote one sentence to that figure. If I understand the result correctly (but see my comment in the 'Minor Issues' section below, noting that I'm not convinced I do), this implies that TPVs grow, reach a maximum size some time in the middle of their life cycle, and then shrink again. This seems interesting and potentially worth digging in to a bit more. Is this expected? Would other tracking methods show a similar result?

Strictly, now Fig. 9 shows that the minimum radius during a TPV's life tends to occur during the beginning or end of the TPV. The reviewer's implied lifecycle is consistent with this pattern. However, so are TPVs that monotonically grow or decay over life. An analogous figure for the normalized lifetime of maximum radius does show that the maximum radius largely occurs between genesis and lysis. This is now noted but not shown.

**Lack of Justification for Methodological Choices**

In general, the authors do a good job of describing their reasoning behind key methodological choices: e.g., the watershed basin method is more robust to 'grid scale undulations' in the field. However, there are a four key choices for which inadequate justification is provided:

1. 'TPVs are defined as tracks with a core at genesis north of 60N lasting at least 2 days' (pg 11, line 7) * TPV minimum latitude: 60N * TPV minimum duration: 2 days

2. 'Default settings are a 300 km filtering disk for regional extrema and 5th percentile of the amplitudes of the basin's boundary cells with respect to the core' (pg 8, lines 16-17) * TPV filtering disk radius: 300 km * TPV percentile threshold: 5th percentile

Of these four choices, the TPV minimum latitude and TPV minimum duration are the least justified: as far as I can tell, the authors simply state this choice without qualification. If another researcher implemented this method, the paper provides no information about why the researcher should conclude that 60N and 2 days should be the default values. Also, these choices effectively embed assumptions about the nature of TPVs in them, and these assumptions (and their implications) should be explicitly stated.

These are the four user-defined parameters in TPVTrack. We reiterate that these are additional degrees of freedom that may impact analyses associated with TPV tracks. Rather than guarantee one set of fixed values, sensitivities should be explored by a user as for any model. We cannot claim that the settings are universally optimal, but they are reasonable for tested and anticipated cases. Rationale for bounds for the parameters have been added to Sect. 2.2.6 Parameter Settings.

Several tradeoffs of the size of the filtering disk and restriction percentile are discussed in the text and illustrated concretely in Figs. 6 and 7. To subset tracks into TPVs associated with the polar vortex that last, 60N is the mean latitude of the polar jet and 2 days matches the definition in H05. These have been added to Sect. 3.2, where the criteria are stated.

The TPV filtering disk radius and TPV percentile threshold do have some explanation provided; the authors appropriately explain that 'Increasing the radius for regional extrema generates larger basins and fewer objects. Increasing the restriction percentile will generate larger basins' (pg 8, lines 18-19). However, the authors also state that 'The default settings best match manual tracks in a small set of case studies.' This seems like a reasonable basis on which to make parameter choices, but what small case studies and which manual tracks? Without this information, a reader has no chance of evaluating whether they agree that the given choices do

result in a good match with case studies and that another choice of parameters would be inferior. I assume that the authors are partly forward referencing the result stated on Pg 12, lines 9-10: "TPVTrack's track exactly matches our manual track," and if so, this should be made explicit. However, this is only one track (the authors indicate that more than one track is used for deciding on parameters), and the track is not actually shown: these manual track are critical data on which the authors are making methodological choices and so should be included in the paper (perhaps as supplementary material?).

The four periods of interest used to inform parameter choices have been listed in Sect. 2.2.6 Parameter Settings.

Since there are dozens of tracks at a given time, comprehensive presentation of all manual maps and TPVTrack tracks (including location and shape) is not simple. Nor is it necessary for the reader's understanding in our opinion as the discussion of the 2006 long-lived track largely covers the pertinent points. Note that the user can easily explore any period of interest by adapting the example ERA-Interim test case and examining the output plots.

**No Discussion of Uncertainties Associated with Methodological Choices**

Related to the above, the authors do not adequately discuss the implications of uncertainties associated with these parameter choices. As noted above, the authors do discuss the effects of varying the filtering disk radius and percentile threshold on individual fields. This is a good direction for discussion, and it would be useful if the authors could expand this discussion to (1) include discussion of implications for changing other parameters, and (2) expand the discussion further to include implications for climate length studies. It would be even more useful if the authors directly showed the effects of these parameter choices on climatological TPV track information.

This is particularly important as there is a growing recognition in the literature that this uncertainty can have major implications for our understanding of weather and climate. For example, the IMILAST project (extratropical cyclone detection intercomparison) shows a ~6x variation in the counts of cyclones across 15 different methods: https://doi.org/10.1175/BAMS-D-11-00154.1. Likewise, ARTMIP (atmospheric river detection intercomparison) a similarly large spread in AR statistics across numerous detection methods (https://journals.ametsoc.org/doi/abs/10.1175/BAMS-D-18-0200.1 and https://www.geosci-model-dev.net/11/2455/2018/gmd-11-2455-2018.html). The ARTMIP project is currently working on experiments to understand whether these different algorithms might produce different climate sensitivities for ARs in climate change experiments. Given this growing understanding in the literature, it is critically important that tracking-method papers such as this explicitly explore uncertainty at the outset. I'm not arguing that the authors should tackle

an intercomparison of the scale of IMILAST or ARTMIP, but since, as the authors note on pg 8 lines 21-22, TPVTrack makes it easy to explore parametric uncertainty, the authors should do just that.

As the reviewer notes, sensitivities to filtering disk radius and percentile threshold are discussed (in combination with varying input data) for the summer 2006 WRF case. To address points (1) and (2), discussion and illustration of the impacts of latitude and lifetime criteria on mean TPV density have been added (end of Sect. 3.2.2 and Fig. 11).

**Minor issues**

pg 1, lines 24-25: "Diagnostic trajectories and prognosed scalar transport further support the advection-dominated dynamics for individual cases (not shown)." <– It is very odd to include a new, not-shown result in the intro: why do this?

We believe that it is important to have context for the expected dynamics of TPVs before considering their tracking. A supporting result comparing the track of the long-lived 2006 TPV with a trajectory model of the core is included in Sect 3.2.1 and referenced in the introduction.

pg 2, lines 4–6: This section should reference IMILAST and ARTMIP, which are both very relevant to the discussion

Reference to ARTMIP has been added. IMILAST was already referenced.

pg 2, lines 18–20: Regarding the first sentence of this paragraph: from where does this qualitative definition originate? If there is a common source (e.g., a textbook), it should be cited. If not, would other polar dynamicists agree on this? Marty Ralph had to convene two AGU townhalls to come to a qualitative, consensus definition of Ars (which is now in the AMS glossary): why would TPVs be different? If this definition is original, I would suggest a rephrasing to make clear that this is a proposed definition:
e.g., "We propose a functional, qualitative definition of TPVs: ..."

The sentence in question states that "Synoptically, TPVs are coherent anomalies on the dynamic tropopause associated with the larger polar vortex that have a regional minimum in potential temperature and cyclonic circulation or regional maximum in potential temperature and anticyclonic circulation that last over time."

In the current AMS glossary's definition of "Polar Vortex" (http://glossary.ametsoc.org/wiki/Polar_vortex; in reference to Cavallo and Hakim 2010), "The term "polar vortex" is sometimes used in reference to smaller-scale (meso- to synoptic scale) vortices that usually occur within the tropospheric polar vortex in polar regions near the

tropopause—for example, "tropopause polar vortices."" Since the manuscript's text is different than the AMS Glossary's partial definition, we have added the suggested text to clarify that this is a proposed definition.

We do not believe that polar dynamicists in consensus would disagree that a planetary polar vortex exists, TPVs are associated and of smaller scale, and TPVs are coherent over time with (nearly-)balanced temperature and wind structure. Of course, the boundaries of a given vortex can vary by definition and possibly should vary by application, as noted in the Conclusions that the "size of a TPV may refer to a number of scales…".

pg 2, lines 20–21: "...are fundamental to an automated scheme" <–I would argue this is true for any objective, quantitative scheme: whether automated or not.

"automated" has been changed to "rule-based" throughout

pg 4, line 6: "through a modular, object-oriented approach is publicly available" <– There seems to be a word missing in this sentence (should it be "*which* is publicly available"?)

The sentence is long but grammatically correct: "An implementation…is publicly available."

pg 8, line 8: 'It is not clear how "optimal" settings would be defined or justified. The default settings best match manual tracks in a small set of case studies.' <– These two sentences seem to contradict each other. The first says we don't know how to define 'optimal', and the second says that we used a small case study to show that our parameter setting results in the best match (which sounds 'optimal' to me).

Rephrased "optimal" to "universally optimal" to clarify that the best settings for all applications are unclear. The case studies have also been listed, which further clarifies the lack of universality.

pg 8, line 30: "...; metrics is independent" ('is' should be 'are')

The modules have been rephrased for grammatical parallelism.

pg 11, line 25: "similar to values found by trapping 2 PVU by searching down from the model top for these grid scales" <– I have no idea what this means. I would suggest rephrasing somehow.

The sentence has been rephrased and now also references Sect. 3.2.3 on diagnosis of the tropopause.

pg 11, line 30: "(Fig 6.e,h,i)" <–Is the lettering here what was actually intended? I'm having a hard time understanding what the authors are referring to.

Added "respectively" to the text. Together with the subplot labels in a later comment, we believe the construction is clearer.

pg 12, lines 9–10: "TPVTrack's track exactly matches our manual track." <–What manual track? I see no figure for this.

Added Fig. 8 showing TPVTrack's track and a map of one time as used for manual tracking.

pg 12, lines 18–19: "Both cyclonic and anticyclonic tracked TPVs reach their minimum radius at the beginning or ends of tracks in the majority of cases (Fig. 8)." I struggled to see how Figure 8 indicates this. It's not that I doubt the result, but rather that the caption for Figure 8 doesn't make sense to me and/or the axis labels are confusing.

The caption has been rephrased.

pg 13, line 25: "...and the bottom of the stratosphere may reach the surface" <– What!? Perhaps there is a polar atmospheric phenomenon that I've not yet learned about, but I've never heard of the tropopause reaching the surface in any dynamical circumstance. I'm wondering if the wording here conveyed something that the authors didn't intend. Otherwise, if this can actually happen, a reference here should be added, since I expect I wouldn't be the only reader to be surprised to learn this.

Reference to a published figure of a cross-section through a PV tower with the tropopause at the surface has been added.

Figure 2b: I read the text and caption several times and I still can't figure out what Figure 2b is supposed to convey.

The caption has been rephrased. Arrows like in 2a have been added for 2b.

Figure 6: Titles/labels on the subplots would be extremely useful. Given that there are 3 resolutions, my initial inclination was to think that columns correspond to resolution– but this isn't true (d,e,f). Because of this confusion, I found I had to repeatedly keep looking between the figures and the caption to understand what I was looking at. It doesn't help that the captions for the subplots reference other parts of the caption ("(i) as in e, but for..."). I found I spent way more time going back and forth between the caption and figures than I normally do in a paper, which made this quite frustrating–and I don't think it needs to be.

The subplots have been labelled with the corresponding grid spacing, closed contour percentile, and filter radius for regional extrema.

Figure 8: I think the caption needs to be reworded. I went back and forth between the figure and the text multiple times before I think I understood the figure. If I understand it correctly, it

might be more usefully worded as "Probability of TPVs being at their minimum equivalent radius as a function of lifecycle for cyclonic...". Also, for the label of the horizontal axis, I would suggest the word 'lifecycle' rather than 'lifetime', because the term lifetime made me think that the horizontal axis refereed to a measure of the duration of the TPV relative to other TPVs.

The caption has been rephrased. Given uncertainty in the stage(s) of a TPV's lifecycle between genesis and lysis, we prefer the term normalized lifetime over lifecycle. This also follows other usage in the literature (e.g., Kew et al. 2010, Fig. 10).

Sincerely,

Nicholas Szapiro and Steven Cavallo

References:

Biernat, K.: Linkages Between Tropopause Polar Vortices and the Development of Cold Air Outbreaks Over Central and Eastern North America, Thesis (M.S.)--State University of New York at Albany, 2017.; Publication Number: AAT 10683411; ISBN: 9780355505092; Source: Masters Abstracts International, Volume: 57-01.; 119 p.

Flaounas, E., Kotroni, V., Lagouvardos, K., and Flaounas, I.: CycloTRACK (v1.0) – tracking winter extratropical cyclones based on relative vorticity: sensitivity to data filtering and other relevant parameters, Geosci. Model Dev., 7, 1841-1853, https://doi.org/10.5194/gmd-7-1841-2014, 2014.

Ullrich, P. A. and Zarzycki, C. M.: TempestExtremes: a framework for scale-insensitive pointwise feature tracking on unstructured grids, Geosci. Model Dev., 10, 1069-1090, https://doi.org/10.5194/gmd-10-1069-2017, 2017.